# Overtuning in Hyperparameter Optimization

**Lennart Schneider**[1,2] **Bernd Bischl**[1,2] **Matthias Feurer**[1,2]

[1]Department of Statistics, LMU Munich
[2]Munich Center for Machine Learning (MCML)

**Abstract** Hyperparameter optimization (HPO) aims to identify an optimal hyperparameter configuration (HPC) such that the resulting model generalizes well to unseen data. As the expected generalization error cannot be optimized directly, it is estimated with a resampling strategy, such as holdout or cross-validation. This approach implicitly assumes that minimizing the validation error leads to improved generalization. However, since validation error estimates are inherently stochastic and depend on the resampling strategy, a natural question arises: Can excessive optimization of the validation error lead to overfitting at the HPO level, akin to overfitting in model training based on empirical risk minimization? In this paper, we investigate this phenomenon, which we term overtuning, a form of overfitting specific to HPO. Despite its practical relevance, overtuning has received limited attention in the HPO and AutoML literature. We provide a formal definition of overtuning and distinguish it from related concepts such as meta-overfitting. We then conduct a large-scale reanalysis of HPO benchmark data to assess the prevalence and severity of overtuning. Our results show that overtuning is more common than previously assumed, typically mild but occasionally severe. In approximately 10% of cases, overtuning leads to the selection of a seemingly optimal HPC with worse generalization error than the default or first configuration tried. We further analyze how factors such as performance metric, resampling strategy, dataset size, learning algorithm, and HPO method affect overtuning and discuss mitigation strategies. Our results highlight the need to raise awareness of overtuning, particularly in the small-data regime, indicating that further mitigation strategies should be studied.

## 1 Introduction

Hyperparameter optimization (HPO) is a fundamental technique in modern machine learning (ML) and allows ML models and complex pipelines to be adapted to different datasets and scenarios (Feurer and Hutter, 2019; Bischl et al., 2023), with pipelines being popular to create full AutoML systems. While resampling-based estimates, such as holdout or cross-validation (CV), are commonly used to construct the objective function in HPO, their stochastic nature can lead to surprising effects on unseen test data (Figure 1). In particular, aggressive optimization of noisy validation scores may result in choosing a hyperparameter configuration (HPC) that performs worse on unseen test data (Ng, 1997; Cawley and Talbot, 2010; Makarova et al., 2021) – a phenomenon we refer to as *overtuning*. Despite its potentially adverse consequences, and although some authors have touched upon this topic in the last 25 years, overtuning has received limited attention in the HPO and AutoML literature and is somewhat underexplored. This paper aims to fill this gap by formally defining overtuning and empirically investigating its prevalence and impact.

Our contributions are as follows: 1) We provide a formal definition of overtuning in HPO, distinguishing it from related concepts such as meta-overfitting and test regret. 2) We reanalyze large-scale HPO benchmark data to quantify how frequently overtuning occurs and assess its practical significance. 3) Through mixed model analyses, we examine how overtuning is influenced by the choice of performance metric, resampling strategy, dataset size, learning algorithm, and HPO method. 4) Finally, we discuss potential mitigation strategies to reduce the risk of overtuning and its extent.

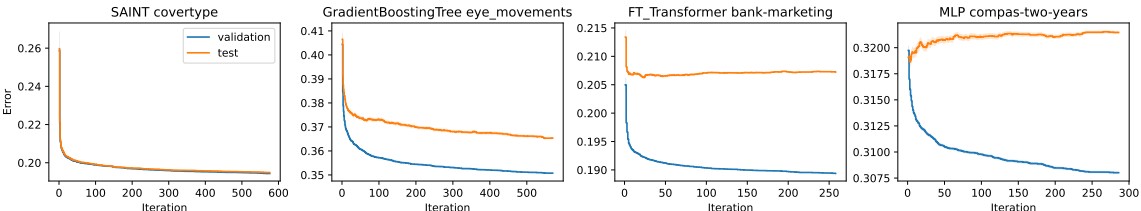

Figure 1: HPO curves based on the data from Grinsztajn et al. (2022). Validation performance of incumbents is given in blue, test performance in orange. From left to right: Ideal, meta-overfitting, benign overtuning, severe overtuning. Ribbons represent standard errors.

## 2  Problem Statement

Background and notation follow Bischl et al. (2023). The goal of supervised ML is to fit a model given $n$ observations, each sampled from a data generating process $\mathbb{P}_{xy}$, so that it generalizes well to new observations from the same data generating process. An ML learning algorithm or inducer $\mathcal{I}$ configured by an HPC $\boldsymbol{\lambda} \in \boldsymbol{\Lambda}$ maps a data set $\mathcal{D}$ to a model $\hat{f}$

$$\mathcal{I} : \mathbb{D} \times \boldsymbol{\Lambda} \to \mathcal{H}, \quad (\mathcal{D}, \boldsymbol{\lambda}) \ \mapsto \hat{f},$$

where $\mathbb{D} := \bigcup_{n \in \mathbb{N}} (\mathcal{X} \times \mathcal{Y})^n$ is the set of all data sets. The search space $\boldsymbol{\Lambda} = \boldsymbol{\Lambda}_1 \times \ldots \times \boldsymbol{\Lambda}_l$ contains all hyperparameters for optimization and their ranges, where $\boldsymbol{\Lambda}_i$ is a bounded subset of the domain of the $i$th hyperparameter. Hyperparameters can be numeric, integer, and categorical. Hierarchical search spaces can arise when the validity of certain hyperparameters depends on the values of others.

In the following, we are concerned with the generalization error (GE) of an inducer $\mathcal{I}$ configured by an HPC $\boldsymbol{\lambda} \in \boldsymbol{\Lambda}$ defined as

$$\mathbb{E}_{\mathcal{D}_{\text{train}} \sim \mathbb{P}_{xy}^n, (\mathbf{x},y) \sim \mathbb{P}_{xy}} \left[ L(y, \mathcal{I}_{\boldsymbol{\lambda}}(\mathcal{D}_{\text{train}})(\mathbf{x})) \right], \tag{1}$$

given a training set $\mathcal{D}_{\text{train}}$ of size $n_{\text{train}}$ and a loss function $L$ with expectation over data set $\mathcal{D}_{\text{train}}$ sampled from $\mathbb{P}_{xy}^n$ and test sample $(\mathbf{x}, y)$ sampled from $\mathbb{P}_{xy}$. To estimate the GE, we make use of a resampling strategy that is denoted as a vector of corresponding splits, i.e., $\mathcal{J} = ((J_{\text{train},1}, J_{\text{test},1}), \ldots, (J_{\text{train},B}, J_{\text{test},B}))$, where $J_{\text{train},i}, J_{\text{test},i}$ are index vectors and $B$ is the number of splits. We then estimate the GE via:

$$\widehat{\text{GE}}(\mathcal{I}, \boldsymbol{\lambda}, \mathcal{J}, L) =$$
$$= \text{agr}\Big( \widehat{\text{GE}}_{J_{\text{train},1}, J_{\text{test},1}}(\mathcal{I}, \boldsymbol{\lambda}, |J_{\text{train},1}|, L), \ldots, \widehat{\text{GE}}_{J_{\text{train},B}, J_{\text{test},B}}(\mathcal{I}, \boldsymbol{\lambda}, |J_{\text{train},B}|, L) \Big). \tag{2}$$

with the aggregator $\text{agr}$ often being chosen as the mean.

Having defined the objective function, we can now state the general HPO problem:

$$\boldsymbol{\lambda}^* \in \operatorname*{arg\,min}_{\boldsymbol{\lambda} \in \boldsymbol{\Lambda}} \widehat{\text{GE}}(\mathcal{I}, \boldsymbol{\lambda}, \mathcal{J}, L). \tag{3}$$

An optimizer sequentially evaluates the ordered sequence of HPCs $(\boldsymbol{\lambda}_1, \ldots \boldsymbol{\lambda}_T)$ with a total budget $T$ – we call such a sequence a "trajectory".[1] The ordered incumbent sequence is $(\boldsymbol{\lambda}_1^*, \ldots, \boldsymbol{\lambda}_T^*)$. Here, each $\boldsymbol{\lambda}_t^*$ is the validation optimal HPC of a trajectory $(\boldsymbol{\lambda}_1, \ldots \boldsymbol{\lambda}_t)$ up to time point $t$:

$$\boldsymbol{\lambda}_t^* := \operatorname*{arg\,min}_{\boldsymbol{\lambda} \in \{\boldsymbol{\lambda}_1, \ldots \boldsymbol{\lambda}_t\}} \widehat{\text{GE}}(\mathcal{I}, \boldsymbol{\lambda}, \mathcal{J}, L).$$

---

[1]We use brackets $(\ldots)$ to denote an ordered sequence, whereas $\{\ldots\}$ denotes an unordered set.

We denote the validation error of an incumbent $\boldsymbol{\lambda}_t^*$ as $\widehat{\mathrm{val}}(\boldsymbol{\lambda}_t^*) := \widehat{\mathrm{GE}}(\mathcal{I}, \boldsymbol{\lambda}_t^*, \mathcal{J}, L)$. We can further denote the true GE of such an optimal $\boldsymbol{\lambda}_t^*$ (fixing the concrete data set $\mathcal{D}_{\mathrm{train}}$ at hand) as:

$$\mathrm{test}(\boldsymbol{\lambda}_t^*) := \mathrm{GE}(\mathcal{I}, \boldsymbol{\lambda}_t^*, \mathcal{D}_{\mathrm{train}}, L) := \mathbb{E}_{(\mathbf{x}, y) \sim \mathbb{P}_{xy}} \left[ L(y, \mathcal{I}_{\boldsymbol{\lambda}_t^*}(\mathcal{D}_{\mathrm{train}})(\mathbf{x})) \right].$$

We can estimate the true GE unbiasedly via another holdout test set or in a nested resampling manner, which we denote by $\widehat{\mathrm{test}}(\boldsymbol{\lambda}_t^*)$.

In this paper we investigate how to quantify the effect that overoptimizing on the validation error may decrease true generalization performance of the incumbent, which we will refer to as the overtuning effect.

## 3 Characterizing the *Overtuning* Effect

Given a sequence of incumbents, $(\boldsymbol{\lambda}_1^*, \ldots, \boldsymbol{\lambda}_t^*)$, we are interested in whether there exists a previous incumbent $\boldsymbol{\lambda}_{t'}^* \in \{\boldsymbol{\lambda}_1^*, \ldots, \boldsymbol{\lambda}_t^*\}$, for which $\mathrm{test}(\boldsymbol{\lambda}_{t'}^*) < \mathrm{test}(\boldsymbol{\lambda}_t^*)$ and, by construction, $\widehat{\mathrm{val}}(\boldsymbol{\lambda}_{t'}^*) \geq \widehat{\mathrm{val}}(\boldsymbol{\lambda}_t^*)$. In other words, have we already observed an incumbent $\boldsymbol{\lambda}_{t'}^*$ that has lower true GE than the actual incumbent $\boldsymbol{\lambda}_t^*$ at time point $t$? And would stopping the HPO process early or choosing the incumbent differently have resulted in lower GE? Based on these questions, we introduce the following definition of overtuning and contrast it with meta-overfitting, trajectory test regret and oracle test regret.

**Definition 3.1.** Given a trajectory $(\boldsymbol{\lambda}_1, \ldots, \boldsymbol{\lambda}_T)$, we define for each time point $1 \leq t \leq T$:

$$\textit{overtuning:} \quad \mathrm{ot}_t(\boldsymbol{\lambda}_1, \ldots, \boldsymbol{\lambda}_t, \ldots, \boldsymbol{\lambda}_T) = \mathrm{test}(\boldsymbol{\lambda}_t^*) - \min_{\boldsymbol{\lambda}_{t'}^* \in \{\boldsymbol{\lambda}_1^*, \ldots, \boldsymbol{\lambda}_t^*\}} \mathrm{test}(\boldsymbol{\lambda}_{t'}^*) \quad (4)$$

$$\textit{meta-overfitting:} \quad \mathrm{of}_t(\boldsymbol{\lambda}_1, \ldots, \boldsymbol{\lambda}_t, \ldots, \boldsymbol{\lambda}_T) = \mathrm{test}(\boldsymbol{\lambda}_t^*) - \widehat{\mathrm{val}}(\boldsymbol{\lambda}_t^*) \quad (5)$$

$$\textit{trajectory test regret:} \quad \mathrm{tr}_t(\boldsymbol{\lambda}_1, \ldots, \boldsymbol{\lambda}_t, \ldots, \boldsymbol{\lambda}_T) = \mathrm{test}(\boldsymbol{\lambda}_t^*) - \mathrm{test}(\boldsymbol{\lambda}_t^\dagger) \quad (6)$$

$$\textit{oracle test regret:} \quad \mathrm{tr}_t(\boldsymbol{\lambda}_1, \ldots, \boldsymbol{\lambda}_t, \ldots, \boldsymbol{\lambda}_T) = \mathrm{test}(\boldsymbol{\lambda}_t^*) - \mathrm{test}(\boldsymbol{\lambda}_t^{\dagger\dagger}) \quad (7)$$

$$\text{where } \boldsymbol{\lambda}_t^\dagger := \operatorname*{arg\,min}_{\boldsymbol{\lambda} \in \{\boldsymbol{\lambda}_1, \ldots, \boldsymbol{\lambda}_t\}} \mathrm{test}(\boldsymbol{\lambda}) \text{ and } \boldsymbol{\lambda}_t^{\dagger\dagger} := \operatorname*{arg\,min}_{\boldsymbol{\lambda} \in \boldsymbol{\Lambda}} \mathrm{test}(\boldsymbol{\lambda}).$$

Overtuning quantifies how much worse the current incumbent performs on true test error compared to the best test-performing incumbent observed so far. In contrast, trajectory test regret compares the current incumbent to all HPCs seen during the search, not just past incumbents. Oracle test regret, on the other hand, quantifies the gap between the current incumbent and the best possible HPC in the entire search space. Since oracle test regret is generally impractical to compute, we refer to trajectory test regret simply as test regret throughout the remainder of the paper. Lastly, meta-overfitting captures the discrepancy between the observed validation error and the true GE, akin to the generalization gap observable on the first level (Hardt and Recht, 2022, Chapter 6). It directly follows that nonzero meta-overfitting is necessary but not sufficient to observe overtuning (see Appendix B), which can also be observed in Figure 1. While investigating meta-overfitting may seem appealing, it is not central to HPO for the following reasons: 1) Validation-test gaps are expected due to finite data and resampling variability. 2) Validation error is mainly used to rank HPCs – its absolute value does not matter. 3) The selected HPC's validation error is a biased estimate of generalization performance anyways. 4) The actual concern in HPO is whether we have selected a seemingly strong HPC that underperforms in true generalization, missing out on a previous better alternative. While overtuning and relative overtuning (Definitions (3.1)–(3.2)) quantify inefficiencies of HPO due to misleading validation signals during optimization, they do not allow for statements regarding absolute generalization performance across different HPO protocols. We illustrate this limitation in Appendix A.

To facilitate comparisons across different tasks, performance metrics, and learning algorithms, we introduce a normalized measure of overtuning. This relative overtuning expresses the overtuning magnitude as a fraction of the maximum possible improvement in test error (with respect to $\boldsymbol{\lambda}_1^* = \boldsymbol{\lambda}_1$ or an explicit default HPC) achieved during the HPO run.

**Definition 3.2.** Given a sequence of HPC evaluations $(\boldsymbol{\lambda}_1, \ldots, \boldsymbol{\lambda}_t, \ldots \boldsymbol{\lambda}_T)$, the relative *overtuning* effect at time point $t$ is defined as

$$\tilde{\mathrm{ot}}_t(\boldsymbol{\lambda}_1, \ldots, \boldsymbol{\lambda}_t, \ldots \boldsymbol{\lambda}_T) = \frac{\mathrm{ot}_t(\boldsymbol{\lambda}_1, \ldots, \boldsymbol{\lambda}_t, \ldots, \boldsymbol{\lambda}_T)}{\mathrm{test}(\boldsymbol{\lambda}_1^*) - \min_{\boldsymbol{\lambda}_{t'}^* \in \{\boldsymbol{\lambda}_1^*, \ldots, \boldsymbol{\lambda}_t^*\}} \mathrm{test}(\boldsymbol{\lambda}_{t'}^*)} \tag{8}$$

Relative overtuning indicates how much worse the current test error is compared to the maximum possible improvement in true generalization performance achieved by HPO. For example, a value of $0$ implies no overtuning, while a value of $0.1$ indicates a $10\%$ loss in test performance made during HPO due to overtuning. Values of $1$ and above imply that overtuning has resulted in no improvement over the initial test error, and we lost all HPO progress and HPO even degraded generalization performance. While relative overtuning quantifies the missed relative improvement due to overtuning, it can overstate the severity of generalization issues when performance gains of HPO are intrinsically small. We discuss this limitation in Appendix A.

## 4 Related Work

We now discuss related work concerned with notions of overtuning in HPO. Appendix C provides an extended discussion and we discuss mitigation strategies in Section 7.

Cawley and Talbot (2010) explore overfitting in model selection, highlighting that criteria like CV estimates of GE have a bias and variance due to finite data. High-variance selection criteria can lead to models that excel on validation data but fail to generalize, an observation consistent with our definition of overtuning, although Cawley and Talbot (2010) do not formally define or quantify it. Their experiments using synthetic data show that validation performance can improve while test performance deteriorates. In contrast, evidence is limited in real-world settings where they observe that a more flexible kernel in kernel ridge regression may overfit validation data compared to a simpler alternative.

Ng (1997) critiques the common practice of selecting models based solely on validation error, noting that the model with the lowest validation error may not have the lowest true GE. This mismatch arises from the variance in the validation error estimator and the sensitivity of the true GE's conditional posterior distribution to the observed validation error conditioned on. This aligns with our definition of overtuning, where validation error may improve while true GE worsens. To address this, Ng (1997) proposes LOOCVCV, which estimates the GE of the best-of-$n$ models for varying $n$ to determine how many models can be considered before overfitting to validation data occurs. The final model is then chosen based on a validation performance percentile $k$ derived from the optimal $n$. On noisy synthetic data, LOOCVCV outperforms naïve selection, but it can be overly conservative in lower-noise settings.

Makarova et al. (2022) propose an early stopping criterion for Bayesian Optimization (BO) in HPO. We refer to Garnett (2023) for a general introduction to BO and to Feurer and Hutter (2019); Bischl et al. (2023) for an introduction in the context of HPO. The early stopping criterion for BO introduced in Makarova et al. (2022) combines a confidence bound on the surrogate model's prediction and the variance of the CV estimator. This approach reduces computational costs with small impact on generalization performance. They also touch on what we define as overtuning, noting that gains in validation performance might not translate to test improvements due to weak validation-test correlations. A prior workshop version (Makarova et al., 2021) highlighted this more explicitly, observing test performance drops in Elastic Net models trained via SGD despite ongoing validation gains.

Lévesque (2018) addresses what we define as overtuning in HPO, showing empirically that validation performance can improve while test performance deteriorates. In a large-scale HPO study tuning support vector machines on 118 datasets using classification error as performance metric, they explore potential mitigation strategies: reshuffling resampling splits, selecting the incumbent on an outer test set (Dos Santos et al., 2009; Koch et al., 2010; Igel, 2013), and selecting the incumbent via the posterior mean in BO. They find that reshuffling improves generalization – especially with holdout as resampling – and that posterior mean selection can further enhance performance. In contrast, additionally holding out a separate selection set harms generalization. While these results support the effectiveness of these strategies, overtuning itself is not formally quantified – its presence is implicitly inferred from improvements in generalization. Nagler et al. (2024) extend this work by demonstrating that reshuffling improves generalization even for a simple random search (RS; Bergstra and Bengio 2012), analyzing its effect on the validation loss surface and deriving regret bounds in the asymptotic regime.

Fabris and Freitas (2019) investigate overfitting in the context of AutoML, conducting experiments with Auto-sklearn (Feurer et al., 2015) on 17 datasets using ROC AUC as the performance metric. They analyze discrepancies across three data partitions: training vs. internal validation, training vs. external test, and internal validation vs. external test – the latter aligning with what we term meta-overfitting. Meta-overfitting is prevalent on smaller datasets (1000 observations or fewer). While validation and test scores are generally well-correlated, the number of HPO iterations by SMAC (Hutter et al., 2011) shows no significant correlation with the extent of meta-overfitting.

In similar spirit, Schröder et al. (2025) focus on the Combined Algorithm Selection and Hyperparameter Optimization (CASH; Thornton et al. 2013) problem and whether meta-overfitting can be observed. They compare RS to BO (SMAC3; Lindauer et al. 2022) on a collection of 64 datasets (binary classification, multiclass classification and regression) and use either holdout or 10-fold CV as the resampling. They differentiate between selection-based meta-overfitting (due to the selection of the final incumbent) and adaptive meta-overfitting (due to the optimizer, such as BO, adapting to the validation loss). They observe that multiclass classification and regression datasets are less affected and that larger validation sets reduce selection-based meta-overfitting but adaptive meta-overfitting can persists on larger sets. Moreover, 10-fold CV reduces meta-overfitting compared to holdout, whereas BO shows higher meta-overfitting than RS with slightly better performance on the outer test set. Moreover, in contrast to Fabris and Freitas (2019) they do observe that the number of HPO iterations correlates positively with adaptive meta-overfitting of BO.

Roelofs et al. (2019) conduct a large-scale empirical study of adaptive overfitting due to test set reuse in Kaggle competitions, finding little evidence of substantial overfitting despite repeated evaluation of models against the public test set. While their setting differs from our explicit focus on HPO, the adaptive dynamics they analyze are related to what we define as meta-overfitting and the resulting optimality gap is implicitly related to what we defined as overtuning.

## 5 An Empirical Analysis of Overtuning

To evaluate the prevalence and practical significance of overtuning in HPO, we re-analyzed several recent, large-scale studies, where the HPO trajectories are publicly available. Specifically, we considered HPO data from the following works: *FCNet* (Klein and Hutter, 2019), *LCBench* (Zimmer et al., 2021), *WDTB* (Grinsztajn et al., 2022), *TabZilla* (McElfresh et al., 2023), *TabRepo* (Salinas and Erickson, 2024), *reshuffling* (Nagler et al., 2024) and *PD1* (Wang et al., 2024). We selected these studies because they include multiple learning algorithms, datasets, and performance metrics. Importantly, each study provides both validation and test performance (estimated on an outer test set), enabling an assessment of overtuning. Each study comprises the evaluation of multiple HPCs for a given combination of learning algorithm, dataset, and performance metric. All studies employed either random search (RS; Bergstra and Bengio 2012) or a fixed grid of HPCs, the latter allowing for simulating a RS. The *reshuffling* study additionally includes BO runs, and runs where

the resampling was reshuffled, and runs where models were not retrained prior to evaluating on the outer test set, which are excluded from the present analysis and revisited in detail in Section 6.

Our empirical analysis aims to answer the questions: 1) How often does overtuning in HPO occur? 2) How strong is the effect? For each HPO run, defined by a unique tuple of learning algorithm, dataset, performance metric, evaluation protocol, and potentially random seed, we computed the relative overtuning as defined in Definition (3.2). Note that the denominator in Equation (8) can cause numerical instabilities. If the default HPC achieves the best test performance over all incumbents or the improvement is small, the denominator will be zero or close to zero, rendering the metric numerically unstable or undefined. Therefore, when quantifying the overtuning effect at a time point $t$, it is reasonable to only consider and average over HPO runs where some improvement over the default can be observed with respect to test performance. We use a threshold of $\epsilon = 0.001$ (with the scale of metrics for, e.g., accuracy and ROC AUC ranging from 0 to 1). This procedure yields a distribution of relative overtuning values per study. Approximately $38.5\%$ of HPO runs yield test performance improvements smaller than this threshold.

We visualize the empirical cumulative distribution function (ECDF) over these values in Figure 2 (solid black line). The analysis reveals that in approximately $60\%$ of HPO runs, no overtuning is observed. Furthermore, $70\%$ of runs exhibit relative overtuning less than $0.1$, while $90\%$ remain below $1.0$. Conversely, this implies that in $10\%$ of HPO runs, we observe what we refer to as "severe" overtuning (i.e., relative overtuning greater than $1.0$). Due to the large variation in the number of HPO runs across studies, we also provide per-study ECDFs in Figure 2. These show substantial heterogeneity: some studies, such as *FCNet*, display almost no overtuning, whereas others, notably *reshuffling* and *TabRepo*, exhibit overtuning in over $50\%$ of runs and severe overtuning in more than $15\%$. We provide additional ECDFs, stratified by learning algorithm, performance metric, and evaluation protocol, for each study in Appendix D and now give a brief summary of key findings.

For *reshuffling* (Figure 3), we observe that across all learning algorithms and performance metrics, overtuning is substantially mitigated when using 5x 5-fold CV, compared to a simple holdout. HPO based on accuracy and ROC AUC tends to result in higher overtuning, whereas log loss is generally more robust. Among the learning algorithms, the Elastic Net displays the lowest sensitivity to overtuning. In contrast, more flexible models as the Funnel MLP, XGBoost and especially CatBoost show substantial overtuning under holdout, although this can be largely alleviated with more sophisticated resamplings. For *WDTB* (Figure 4), we observed that overtuning is most pronounced for classification tasks evaluated using accuracy, particularly on the categorical classification benchmarks. In contrast, numerical regression tasks using $R^2$ exhibit substantially lower overtuning. Among learning algorithms, tree-based models such as GradientBoostingTree and HistGradientBoostingTree demonstrate the greatest robustness. Neural architectures, particularly the ResNet and MLP show higher overtuning, especially on classification tasks. Looking at *TabZilla* (Figures 5, 6, 7, 8), we observed that the tree-based gradient boosting algorithms are relatively robust to overtuning, particularly on multiclass classification. Neural architectures including the ResNet and especially the MLPs are more prone to overtuning. In general, binary classification (ROC AUC) is more sensitive to overtuning than multiclass classification (log loss). This is consistent with theoretical results showing that overfitting due to test set reuse is harder in multiclass settings with many classes (Feldman et al., 2019). For TabRepo (Figure 9), we observed the similar trend that binary classification (ROC AUC) is more sensitive to overtuning than multiclass classification (log loss) or regression (RMSE). Moreover, CatBoost and the two neural architectures are more prone to overtuning than the other learning algorithms. For LCBench (Figure 10), PD1 (Figure 11), and FCNet (Figure 12), we observed minimal overtuning but noticed that accuracy or classification error are more sensitive to overtuning than cross-entropy, i.e., log loss.

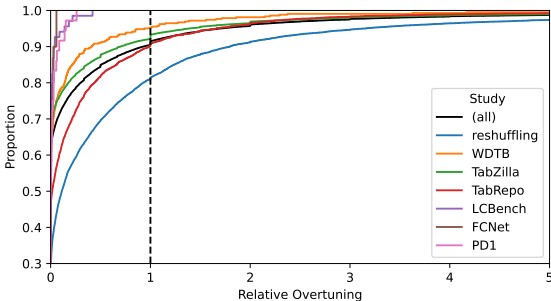

Figure 2: ECDFs of relative overtuning over different HPO studies. $y$-axis starts at 0.3.

## 6 Modeling the Determinants of Overtuning

To directly investigate overtuning in HPO and identify influential factors, such as learning algorithms, performance metrics, evaluation protocols, and optimizers, we analyze the *reshuffling* HPO data (Nagler et al., 2024) in more detail. In that study, the authors systematically varied the learning algorithm (Elastic Net (Zou and Hastie, 2005), Funnel MLP (Zimmer et al., 2021), XGBoost (Chen and Guestrin, 2016), CatBoost (Prokhorenkova et al., 2018)), performance metric (accuracy, log loss, ROC AUC), dataset size ($n = 500, 1000$, or $5000$ observations, with a fixed outer test set of size 5000), and resampling strategy (80/20 holdout, 5-fold CV, 5x 80/20 holdout, 5x 5-fold CV) in a full factorial design, repeating each HPO run on ten binary classification datasets (treated as data generating processes) ten times. For our analysis, we focus on results from RS with a budget of 500 HPC evaluations under default non-reshuffled resampling, comprising 14400 HPO runs in total. Test performance was determined by retraining the inducer configured by a given HPC on all data and evaluating on the outer holdout set. For more details, see Nagler et al. (2024).

We investigate how overtuning is influenced by the number of HPO iterations, performance metric, learning algorithm (classifier), resampling strategy, and dataset size. Rather than testing strict hypotheses, our analysis is exploratory (Herrmann et al., 2024). Overtuning and relative overtuning are computed per HPO run as defined in Definitions (3.1)–(3.2). Since many runs show no overtuning, we first fit a generalized linear mixed-effects model (GLMM) to predict the probability of nonzero overtuning. The model includes random intercepts for dataset and seed, and fixed effects for the performance metric, classifier, resampling strategy, dataset size, and a scaled HPO budget (0 to 1), including a quadratic term for the budget to capture nonlinearity. We omit interaction terms to keep the model simple. Results are shown in Table 1a. We observe that longer tuning increases the odds of overtuning (positive main effect), but the negative quadratic term indicates a diminishing effect at higher iteration counts, forming a plateau similar to an inverted U-shape. A likelihood ratio test confirms the necessity of the quadratic term ($\chi^2(1) = 1913.90, p < 0.001$). Compared to the reference levels (accuracy for metric and Elastic Net for classifier), both log loss and ROC AUC increase the odds of overtuning, and all classifiers increase these odds. In contrast, employing more sophisticated resampling strategies (especially 5x 5-fold CV compared to holdout) and using more data ($n = 1000$ or $n = 5000$ observations instead of $n = 500$) reduces the odds.

As a follow up, we fitted a linear mixed-effects model (LMM) to predict the relative overtuning as in Definition (3.2) on a logarithmic scale (to counter skewness) for cases with nonzero overtuning. The LMM uses the same random and fixed effects as the GLMM, and a likelihood ratio test again confirm the need for a quadratic budget term ($\chi^2(1) = 31.361, p < 0.001$). Table 1b summarizes these results. The conclusions largely remain the same as for the GLMM, i.e., using a more sophisticated resampling strategy and more data reduces the extent of overtuning although ROC AUC and log loss now overall show less nonzero relative overtuning compared to accuracy. As before, longer tuning increases relative overtuning (positive main effect), but the negative quadratic

(a) Fixed effects results from a GLMM predicting probability of nonzero overtuning.

| Predictor | Estimate | Std. Error | z value | p-value |
|---|---|---|---|---|
| (intercept) | -1.408569 | 0.096734 | -14.560 | < 0.001 |
| budget | 2.262315 | 0.035965 | 62.900 | < 0.001 |
| budget$^2$ | -1.495099 | 0.034100 | -43.840 | < 0.001 |
| metric (ROC AUC) | 0.724608 | 0.006277 | 115.440 | < 0.001 |
| metric (log loss) | 0.222283 | 0.006186 | 35.930 | < 0.001 |
| classifier (CatBoost) | 1.609866 | 0.007494 | 214.810 | < 0.001 |
| classifier (Funnel MLP) | 1.200907 | 0.007357 | 163.230 | < 0.001 |
| classifier (XGBoost) | 1.336699 | 0.007390 | 180.890 | < 0.001 |
| resampling (5x Holdout) | -0.264233 | 0.007202 | -36.690 | < 0.001 |
| resampling (5-fold CV) | -0.290060 | 0.007202 | -40.270 | < 0.001 |
| resampling (5x 5-fold CV) | -0.481657 | 0.007214 | -66.770 | < 0.001 |
| dataset size (1000) | -0.275075 | 0.006229 | -44.160 | < 0.001 |
| dataset size (5000) | -0.640027 | 0.006261 | -102.230 | < 0.001 |

(b) Fixed effects results from an LMM predicting nonzero relative overtuning on log scale.

| Predictor | Estimate | Std. Error | df | t value | p-value |
|---|---|---|---|---|---|
| (intercept) | -1.962e+00 | 1.465e-01 | 1.809e+01 | -13.389 | < 0.001 |
| budget | 3.955e-01 | 4.089e-02 | 2.616e+05 | 9.672 | < 0.001 |
| budget$^2$ | -2.093e-01 | 3.737e-02 | 2.616e+05 | -5.600 | < 0.001 |
| metric (ROC AUC) | -2.178e-01 | 6.922e-03 | 2.616e+05 | -31.463 | < 0.001 |
| metric (log loss) | -7.852e-01 | 7.167e-03 | 2.616e+05 | -109.548 | < 0.001 |
| classifier (CatBoost) | 2.693e+00 | 8.887e-03 | 2.616e+05 | 302.994 | < 0.001 |
| classifier (Funnel MLP) | 1.218e+00 | 8.855e-03 | 2.616e+05 | 137.563 | < 0.001 |
| classifier (XGBoost) | 2.176e+00 | 9.235e-03 | 2.616e+05 | 235.615 | < 0.001 |
| resampling (5x Holdout) | -3.165e-01 | 7.390e-03 | 2.616e+05 | -42.831 | < 0.001 |
| resampling (5-fold CV) | -3.081e-01 | 7.371e-03 | 2.616e+05 | -41.793 | < 0.001 |
| resampling (5x 5-fold CV) | -4.927e-01 | 7.530e-03 | 2.616e+05 | -65.437 | < 0.001 |
| dataset size (1000) | -1.291e-01 | 6.285e-03 | 2.616e+05 | -20.549 | < 0.001 |
| dataset size (5000) | -4.136e-01 | 6.658e-03 | 2.616e+05 | -62.129 | < 0.001 |

Table 1: Fixed effects results of mixed models used to analyze overtuning. RS runs, no reshuffling, test performance of the model retrained on all data. Reference levels: accuracy (metric), Elastic Net (classifier), holdout (resampling), 500 (dataset size).

(a) Fixed effects results from an LMM predicting final meta-overfitting.

| Predictor | Estimate | Std. Error | df | t value | p-value |
|---|---|---|---|---|---|
| (intercept) | 3.016e-02 | 3.803e-03 | 1.076e+01 | 7.931 | < 0.001 |
| metric (ROC AUC) | 2.148e-02 | 7.691e-04 | 1.437e+04 | 27.930 | < 0.001 |
| metric (log loss) | -3.619e-03 | 7.691e-04 | 1.437e+04 | -4.705 | < 0.001 |
| classifier (CatBoost) | 2.064e-02 | 8.880e-04 | 1.437e+04 | 23.242 | < 0.001 |
| classifier (Funnel MLP) | 1.736e-02 | 8.880e-04 | 1.437e+04 | 19.545 | < 0.001 |
| classifier (XGBoost) | 1.154e-02 | 8.880e-04 | 1.437e+04 | 12.998 | < 0.001 |
| resampling (5x Holdout) | -1.733e-02 | 8.880e-04 | 1.437e+04 | -19.514 | < 0.001 |
| resampling (5-fold CV) | -2.022e-02 | 8.880e-04 | 1.437e+04 | -22.769 | < 0.001 |
| resampling (5x 5-fold CV) | -2.830e-02 | 8.880e-04 | 1.437e+04 | -31.868 | < 0.001 |
| dataset size (1000) | -1.281e-02 | 7.691e-04 | 1.437e+04 | -16.661 | < 0.001 |
| dataset size (5000) | -2.562e-02 | 7.691e-04 | 1.437e+04 | -33.317 | < 0.001 |

(b) Fixed effects results from an LMM predicting final test regret.

| Predictor | Estimate | Std. Error | df | t value | p-value |
|---|---|---|---|---|---|
| (intercept) | 1.082e-02 | 1.333e-03 | 1.305e+01 | 8.120 | < 0.001 |
| metric (ROC AUC) | 1.154e-02 | 4.157e-04 | 1.437e+04 | 27.754 | < 0.001 |
| metric (log loss) | -1.240e-04 | 4.157e-04 | 1.437e+04 | -0.298 | < 0.001 |
| classifier (CatBoost) | 6.822e-03 | 4.800e-04 | 1.437e+04 | 14.212 | < 0.001 |
| classifier (Funnel MLP) | 1.215e-02 | 4.800e-04 | 1.437e+04 | 25.321 | < 0.001 |
| classifier (XGBoost) | 4.122e-03 | 4.800e-04 | 1.437e+04 | 8.587 | < 0.001 |
| resampling (5x Holdout) | -5.437e-03 | 4.800e-04 | 1.437e+04 | -11.327 | < 0.001 |
| resampling (5-fold CV) | -5.839e-03 | 4.800e-04 | 1.437e+04 | -12.164 | < 0.001 |
| resampling (5x 5-fold CV) | -7.362e-03 | 4.800e-04 | 1.437e+04 | -15.338 | < 0.001 |
| dataset size (1000) | -5.352e-03 | 4.157e-04 | 1.437e+04 | -12.876 | < 0.001 |
| dataset size (5000) | -1.027e-02 | 4.157e-04 | 1.437e+04 | -24.696 | < 0.001 |

Table 2: Fixed effects results of mixed models used to analyze final meta-overfitting and test regret. RS runs, no reshuffling, test performance of the model retrained on all data. Reference levels of factors are: accuracy (metric), Elastic Net (classifier), holdout (resampling), 500 (dataset size).

term indicates a diminishing effect. Finally, we fitted LMMs to predict the final meta-overfitting and final test regret after 500 HPO iterations. Results in Table 2a and Table 2b show that employing more sophisticated resampling strategies and using larger datasets reduce both meta-overfitting and test regret. These findings suggest that practitioners should prefer CV (repeated if possible) over holdout validation whenever possible, particularly with small datasets.

To assess the effect of the optimizer (RS vs. HEBO, see Cowen-Rivers et al. 2022 vs. SMAC3, see Lindauer et al. 2022), we conduct another mixed model analysis on the reshuffling data subset, limited to 250 iterations (the BO budget), using ROC AUC as the performance metric (the only one tracked in BO experiments). The choice of optimizer is included as a fixed effect and other random and fixed effects remain the same as in the previous modeling approach. A likelihood ratio test reveals a significant effect of the optimizer for both the GLMM modeling the probability of nonzero overtuning and the LMM modeling the nonzero relative overtuning on log scale: $\chi^2(2) = 416.14, p < 0.001$ for the GLMM, and $\chi^2(2) = 1509.7, p < 0.001$ for the LMM. In the GLMM (Table 3a), we observed small but significant positive coefficients for both HEBO ($0.0833, z = 9.049, p < 0.001$) and SMAC3 ($0.1881, z = 20.347, p < 0.001$), compared to RS, suggesting that both BO methods slightly increase the odds of nonzero overtuning. Conversely, the LMM analysis of the magnitude of overtuning (Table 3b) shows significant negative coefficients for HEBO ($-0.3011, t(154200) = -36.363, p < 0.001$) and SMAC ($-0.2581, t(154200) = -30.956, p < 0.001$), indicating that while BO slightly increases the likelihood of any overtuning, it substantially reduces its magnitude compared to RS. Finally, based on an LMM modeling final test regret with optimizer as a fixed factor (Table 4b), we find that HEBO significantly reduces test regret relative to RS ($-0.0021, t(14370) = -5.167, p < 0.001$), suggesting that HEBO tends to identify HPCs that generalize better. SMAC3 also shows a small negative coefficient ($-0.0003, t(14370) = -0.691, p = 0.489$), but this effect is not statistically significant.

We also investigate the effect of early stopping in BO (Makarova et al., 2021, 2022) by comparing HEBO with HEBO using early stopping on the data subset up to 250 iterations (the BO budget), using 5-fold CV as the resampling strategy (the only setting where early stopping à la Makarova et al. (2022) is directly applicable), and ROC AUC as the performance metric (the only one tracked in BO experiments). We apply the same mixed model analysis framework as before. Likelihood ratio tests reveal a significant effect of early stopping for both the probability of nonzero overtuning (GLMM: $\chi^2(1) = 36.077$, $p < 0.001$) and the extent of nonzero relative overtuning on log scale (LMM: $\chi^2(1) = 10.720$, $p = 0.001$). When including early stopping as a fixed factor (Table 5b), we observed a negative coefficient ($-0.27531$, $t(980.555) = -3.272$, $p = 0.001$) for nonzero relative overtuning on log scale indicating a mitigating effect, albeit comparably small. One reason can be that this analysis is restricted to HPO runs using 5-fold CV, where we have seen that overtuning is rather mild, when compared to holdout runs. Moreover, since HEBO already reduces overtuning compared to RS, the additional benefit from applying early stopping can be rather incremental.

Last but not least, we turn to the core idea behind the reshuffling data: reshuffling the resampling splits during HPO, a strategy shown to improve generalization performance, particularly in the case of holdout resampling (Nagler et al., 2024). We conduct a mixed model analysis as before on the reshuffling data, focusing on the larger subset of RS runs (500 iterations). A likelihood ratio test indicates a significant effect of reshuffling for both the GLMM modeling the probability of nonzero overtuning ($\chi^2(1) = 152.54$, $p < 0.001$) and the LMM modeling the nonzero relative overtuning on log scale ($\chi^2(1) = 181.10$, $p < 0.001$). Specifically, we find that, overall, reshuffling slightly increases the odds of overtuning ($0.0439$, $z = 12.351$, $p < 0.001$, Table 6a) as well as its extent ($0.0515$, $t(537900) = 13.458$, $p < 0.001$, Table 6b). Nagler et al. (2024) demonstrated that reshuffling can improve generalization especially when holdout is used as a resampling with ROC AUC as the performance metric. When we restrict our analysis to this particular setting, we observed a clear shift: For both the GLMM (Table 8a) and LMM (Table 8b), reshuffling has a significant negative effect on overtuning: it strongly decreases the odds of overtuning ($-0.2645$, $z = -20.054$, $p < 0.001$) and its extent ($-0.2693$, $t(55480) = -23.236$, $p < 0.001$). Moreover, reshuffling significantly reduces final test regret in this setting, as shown by an LMM analysis ($-0.0057$, $t(2375) = -5.990$, $p < 0.001$, Table 9b), indicating that it leads to the identification of HPCs with better true generalization performance. We find that reshuffling actually increases final meta-overfitting ($0.0548$, $t(2375) = 25.382$, $p < 0.001$, Table 9a). However, meta-overfitting does not necessarily imply worse HPO generalization. In fact, the "hedging"' effect of reshuffling described by Nagler et al. (2024) appears strong enough to reduce both overtuning and test regret.

## 7 Mitigation Strategies

While the primary contribution of this paper is to highlight the issue of overtuning in HPO, we now turn to a discussion of potential mitigation strategies, drawing from both Section 6 and existing literature. Broadly, these strategies fall into three categories: 1) Modifying the objective function, 2) adjusting incumbent selection (either already during optimization or only as a simple post-hoc step), and 3) modifying the optimizer producing the HPC trajectory. The first category includes methods that reduce variance (e.g., by using a more sophisticated resampling strategy) or add regularization. The second category includes early stopping and avoiding naïve selection of the validation-optimal HPC. The third category is naturally broader and includes any changes to the optimizer that produces the HPC trajectory.

We have seen in our analysis of the data from Nagler et al. (2024) that larger datasets generally reduce both the likelihood and severity of overtuning. While this is expected, we note that increasing dataset size is often infeasible in practice. As such, overtuning remains to a large extent a primary concern in small-data regimes. Moreover, more advanced resampling strategies such as CV or repeated CV substantially reduce both the frequency and extent of overtuning. These strategies, along with larger datasets, also help mitigate meta-overfitting and decrease test regret,

enabling HPO to more reliably identify configurations that generalize well. Additionally, our findings suggest that BO results in less overtuning than RS, although it may slightly increase meta-overfitting. This trade-off deserves further study. One possible explanation is that BO more effectively identifies configurations with exceptionally strong validation performance that also generalize well. Besides, BO's use of a surrogate model may help smooth over noise in validation estimates, guiding the search toward more robust well-performing regions. In noisy BO, one can select the (final) incumbent based on the surrogate's posterior predictive distribution rather than the empirically best configuration (Picheny et al., 2013). While this was not implemented in the BO runs of Nagler et al. (2024), incorporating such noise-aware techniques may further reduce overtuning as briefly touched upon by Lévesque (2018). Finally, reshuffling resampling splits as done in Lévesque (2018); Nagler et al. (2024) can help mitigate overtuning, although its effectiveness varies across performance metrics, algorithms, and resampling strategies.

Prior work has touched on several strategies to mitigate overtuning. Cawley and Talbot (2010) briefly mention regularization, early stopping, and model averaging. For example, Cawley and Talbot (2007) show that incorporating L2 regularization on lengthscale parameters in kernel methods can improve generalization. However, in modern tabular learning settings, applying regularization during HPO is challenging, as it requires a clear mapping between hyperparameters and model complexity – something not always available. Early stopping, explored by Makarova et al. (2021, 2022), shows promise, but in our analysis, it did not strongly reduce overtuning. One reason could be that stopping too early may prevent discovering genuinely better configurations. Another issue lies in the reliability of the variance estimator used for the stopping criterion, where we know that no unbiased variance estimator exists for CV performance estimates (Bengio and Grandvalet, 2004). Finally, a fully Bayesian treatment of hyperparameters, as presented in Williams and Barber (1998) and discussed by Cawley and Talbot (2010), appears impractical for modern models due to computational and modeling complexity.

Other mitigation strategies focus on more cautious incumbent selection, such as using a dedicated selection set or applying conservative selection criteria. Several works (Dos Santos et al., 2009; Koch et al., 2010; Igel, 2013; Lévesque, 2018) have explored selecting the final HPC based on a separate test set. However, Lévesque (2018) report that a dedicated test set can degrade generalization performance, as it reduces the data available for HPO. Similarly, ML-Plan (Mohr et al., 2018) adopts a two-phase strategy: It first explores candidates using one cross-validation on a subset of the training data split, and then re-evaluates the best HPCs on the full training data set (after shuffling it once). LOOCVCV (Ng, 1997) proposes selecting the incumbent not with the best validation error, but at an adaptively chosen percentile, based on how many configurations can be evaluated before overtuning occurs. While effective in noisy settings, it tends to be overly conservative in low-noise regimes and is limited to decomposable point-wise metrics and i.i.d. configurations, restricting it to RS while adding computational overhead.

We have seen that an effective and simple mitigation strategy against overtuning is using more robust resampling strategies like repeated CV. However, this comes with increased computational cost. To balance robustness and efficiency, it may be worthwhile to revisit adaptive resampling techniques (Thornton et al., 2013; Zheng and Bilenko, 2013; Bergman et al., 2024; Buczak et al., 2024), racing (Birattari et al., 2002; Lang et al., 2015) or optimal computing budget allocation strategies (Bartz-Beielstein et al., 2011) that all stop the evaluation of likely poor-performing HPCs early.

## 8 Broader Impact Statement

This work presents no notable negative impacts to society or the environment.

**Acknowledgements**. We thank Ricardo Knauer for pointers to applied work on overtuning of linear models.

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

## A   Limitations

Overtuning and relative overtuning, as defined in Definition (3.1) and Definition (3.2), quantify how much better HPO could have performed if no decisions had been made based on misleading validation error throughout the trajectory of evaluated HPCs. However, these metrics are not designed to assess or compare the absolute generalization performance of different HPO protocols. While this may seem evident, we state it explicitly for clarity. Consider two hypothetical HPO protocols:

- Protocol A evaluates only a single configuration, $\boldsymbol{\lambda}_{A,1} = \boldsymbol{\lambda}_{A,1}^*$, achieving $\widehat{\mathrm{val}}(\boldsymbol{\lambda}_{A,1}^*) = 0.3$ and $\mathrm{test}(\boldsymbol{\lambda}_{A,1}^*) = 0.35$.

- Protocol B in contrast evaluates $T = 10$ configurations, with the final incumbent $\boldsymbol{\lambda}_{B,10}^*$ achieving $\widehat{\mathrm{val}}(\boldsymbol{\lambda}_{B,10}^*) = 0.18$ and $\mathrm{test}(\boldsymbol{\lambda}_{B,10}^*) = 0.22$.

Suppose Protocol B exhibits a final overtuning of $\mathrm{ot}_{10} = 0.02$, implying that an earlier incumbent had a true GE of 0.20. Protocol A, by definition, cannot exhibit overtuning since it only evaluates a single configuration. Nevertheless, Protocol B clearly leads to better generalization performance (0.22 vs. 0.35), and should be preferred when the primary concern is generalization – even though it exhibits overtuning. However, the presence of overtuning in Protocol B is still informative, as it indicates that even better generalization performance was theoretically achievable.

Furthermore, relative overtuning (Definition (3.2)) can be sensitive to the scale of possible performance improvements. If the test error difference between the default or first evaluated HPC and the best observed incumbent is small, the relative overtuning may appear disproportionately large. This reflects the metric's design – to measure the relative gain missed due to overtuning – but it can lead to inflated values in scenarios where HPO yields only marginal improvements. We do not investigate in this work in detail why such marginal improvements might occur and how improvements on the validation set can better generalize to improvements on an outer test set but note that meta-overfitting (Definition (3.1)) can be a suitable metric to assess this. Possible explanations include low tunability (Probst et al., 2019; van Rijn and Hutter, 2018) of the learning algorithm, or overly constrained search spaces where all configurations perform similarly. In such cases, high relative overtuning may simply reflect the limited room for improvement rather than poor HPO generalization.

A direct practical implication is that overtuning alone is not sufficient to evaluate or compare HPO protocols. When analyzing mitigation strategies for overtuning, it is essential to consider their impact on absolute generalization performance. In specific settings – such as the RS runs in Nagler et al. (2024), where all protocols use the same fixed trajectory of HPCs – trajectory test regret (Definition (3.1)) may suffice, as it directly measures how well a protocol identifies a near-optimal configuration within the same trajectory. However, when HPO protocols differ in their search trajectories (and their length) – due to early stopping, different optimizers, or resource budgets – we must also compare the final test performance of their incumbents. Only then can we draw reliable conclusions about which protocol performs better overall with respect to generalization. For the HEBO vs. HEBO with early stopping à la Makarova et al. (2022) analyses reported in Section 6, we therefore provide a follow-up analysis concerned with the test performance of the final incumbent in Appendix E.1.

## B Overtuning vs. Meta-Overfitting

**Proposition B.1.** *Given a sequence of HPC evaluations $(\boldsymbol{\lambda}_1, \ldots, \boldsymbol{\lambda}_t, \ldots \boldsymbol{\lambda}_T)$, if overtuning exists at a time point $t$, i.e., $\mathrm{ot}_t(\boldsymbol{\lambda}_1, \ldots, \boldsymbol{\lambda}_t, \ldots \boldsymbol{\lambda}_T) > 0$, there must exist some nonzero meta-overfitting for the current incumbent $\boldsymbol{\lambda}_t^*$ or some previous incumbent $\boldsymbol{\lambda}_{t'}^*$ $(t' < t)$, i.e., $\mathrm{of}_t(\boldsymbol{\lambda}_1, \ldots, \boldsymbol{\lambda}_t, \ldots \boldsymbol{\lambda}_T) \neq 0$ or $\mathrm{of}_{t'}(\boldsymbol{\lambda}_1, \ldots, \boldsymbol{\lambda}_t, \ldots \boldsymbol{\lambda}_T) \neq 0$.*

We can easily see this by contradiction. Assume that overtuning exists at time point $t$. By Definition (3.1) this means $(\mathrm{test}(\boldsymbol{\lambda}_t^*) - \min_{\boldsymbol{\lambda}_{t'}^* \in \{\boldsymbol{\lambda}_1^*, \ldots, \boldsymbol{\lambda}_t^*\}} \mathrm{test}(\boldsymbol{\lambda}_{t'}^*)) > 0$. Since the minimum is strictly less than $\mathrm{test}(\boldsymbol{\lambda}_t^*)$ there must exist a previous incumbent $\boldsymbol{\lambda}_{t'}^*$ such that $\mathrm{test}(\boldsymbol{\lambda}_{t'}^*) < \mathrm{test}(\boldsymbol{\lambda}_t^*)$. By definition of incumbents, we know: $\widehat{\mathrm{val}}(\boldsymbol{\lambda}_t^*) \leq \widehat{\mathrm{val}}(\boldsymbol{\lambda}_{t'}^*)$ since $t' < t$. Assume that meta-overfitting is zero for both $\boldsymbol{\lambda}_t^*$ and $\boldsymbol{\lambda}_{t'}^*$, i.e., $\mathrm{test}(\boldsymbol{\lambda}_t^*) = \widehat{\mathrm{val}}(\boldsymbol{\lambda}_t^*)$ and $\mathrm{test}(\boldsymbol{\lambda}_{t'}^*) = \widehat{\mathrm{val}}(\boldsymbol{\lambda}_{t'}^*)$. Substituting in $\mathrm{test}(\boldsymbol{\lambda}_{t'}^*) < \mathrm{test}(\boldsymbol{\lambda}_t^*)$ gives $\widehat{\mathrm{val}}(\boldsymbol{\lambda}_{t'}^*) = \mathrm{test}(\boldsymbol{\lambda}_{t'}^*) < \mathrm{test}(\boldsymbol{\lambda}_t^*) = \widehat{\mathrm{val}}(\boldsymbol{\lambda}_t^*)$, from which follows $\widehat{\mathrm{val}}(\boldsymbol{\lambda}_{t'}^*) < \widehat{\mathrm{val}}(\boldsymbol{\lambda}_t^*)$ contradicting the established relation $\widehat{\mathrm{val}}(\boldsymbol{\lambda}_t^*) \leq \widehat{\mathrm{val}}(\boldsymbol{\lambda}_{t'}^*)$.

Note that nonzero meta-overfitting, however, is not sufficient to observe overtuning. Assume the following performance values of HPCs $\boldsymbol{\lambda}_1, \boldsymbol{\lambda}_2$: $\widehat{\mathrm{val}}(\boldsymbol{\lambda}_1) = 0.3, \widehat{\mathrm{val}}(\boldsymbol{\lambda}_2) = 0.2, \mathrm{test}(\boldsymbol{\lambda}_1) = 0.4, \mathrm{test}(\boldsymbol{\lambda}_2) = 0.35$. we observed meta-overfitting of $\mathrm{of}_1(\boldsymbol{\lambda}_1, \boldsymbol{\lambda}_2) = 0.4 - 0.3 = 0.1$ and $\mathrm{of}_2(\boldsymbol{\lambda}_1, \boldsymbol{\lambda}_2) = 0.35 - 0.2 = 0.15$. Still, overtuning is zero as neither for $t = 1$ nor $t = 2$ there exists a previous incumbent with better test performance. In this sense, we correctly identified the best HPC performing with respect to true GE. This relationship of meta-overfitting and overtuning is also depicted in Figure 1. Naturally, if meta-overfitting is simply a gap between validation and test error that is (roughly) the same for all incumbents or HPCs, there cannot be any overtuning.

## C  Extended Related Work

A foundational study by Cawley and Talbot (2010) shows that any model selection criterion (for instance, CV) inherently has both bias and variance because it relies on a finite data sample. As they illustrate with synthetic data, extensive optimization on the validation set can cause the chosen model to excel on validation performance but fail to generalize to unseen test data. Their real-data experiments focus on comparing final configurations chosen by different model-selection schemes (e.g., a single-parameter RBF kernel vs. an ARD kernel for kernel ridge regression). They do not define a formal metric of overfitting for model selection but empirically demonstrate that more flexible setups, such as ARD, can outperform on validation yet yield worse performance on a held-out test set. Their work is best known for emphasizing nested CV (or nested resampling in general) as essential for unbiased performance estimation once model selection is performed.

Ng (1997) propose an approach closely aligned with the idea of overtuning, although their work has not been widely recognized in contemporary AutoML and HPO research. They critique the practice of selecting models solely by their CV (in actuality, simple holdout; earlier works used the term CV for holdout, and k-fold CV for what is nowadays meant by CV) performance, noting that the variance of the validation error estimate can skew the posterior distribution of the true GE. To address this, Ng (1997) suggests selecting not the lowest validation error, but rather the hypothesis at the $k$-th percentile of validation performance, where $k$ is chosen adaptively. This adaptation relies on LOOCVCV: it estimates how many candidate hypotheses can be evaluated before overfitting to the validation error. Although effective under high noise and limited samples (e.g., in synthetic classification with decision trees), this LOOCVCV approach can become overly conservative with lower noise, sometimes underperforming simpler selection strategies.

Guyon et al. (2010) further formalize model selection through a multi-level inference framework that brings together Bayesian, frequentist, and hybrid viewpoints (see also Bischl et al. 2023). They underscore the risk of overfitting in hyperparameter selection – citing Cawley and Talbot (2010) – and advocate for bound-based selection, ensemble methods, and other regularization techniques to mitigate it. Likewise, Guyon et al. (2015) view model selection as a bi-level optimization problem, arguing that one must introduce regularization and robust data-splitting practices to avoid overfitting to empirical criteria like the CV error. Auto-sklearn 2.0 (Feurer et al., 2022) demonstrate that it is also possible to meta-learn the model selection criterion rather than treating it as a static heuristic.

Makarova et al. (2022) address the challenge of deciding when to stop BO in HPO by proposing a new termination criterion. This criterion combines a confidence bound on the surrogate model's regret with a variance estimate of the CV estimator. Their rule halts BO once the maximum plausible improvement from the surrogate falls below the standard deviation of the incumbent's validation error. They report that this avoids many unnecessary function evaluations and saves computational resources, at only a small cost in final test performance. While they briefly acknowledge that the discrepancy between validation and test performance can persist, it is attributed mainly to low validation–test correlation.

An earlier workshop version (Makarova et al., 2021) puts stronger emphasis on "overfitting" in BO, showing that in tuning an Elastic Net, XGBoost, and a random forest across 19 datasets, Elastic Net (trained via SGD) performance on the test set often declined after prolonged validation-driven optimization. Their explanation again points to weak validation–test correlations, though they do not discuss deeper causes (dataset traits, algorithms, metrics, or resampling choices). In their analyses, they employ the Relative Test Error Change (RYC) to compare test errors in early-stopped vs. full-budget runs, and the Relative Time Change (RTC) to quantify computational savings. Positive RYC implies that early stopping helped avert overtuning, whereas negative values mean the run was halted prematurely.

Nguyen et al. (2018) also study overfitting in BO-based HPO, focusing on how to detect "stable" solutions. Their notion of stability involves low "extra variance", defined as the change in predictive mean and variance under small Gaussian perturbations of the hyperparameters. A high extra variance signals a rapidly varying objective function that may lead to overtuning. They propose two stability-aware acquisition functions, Stable-UCB and Stable-EI, which penalize instability to encourage more robust HPCs.

Other works on early stopping in BO include Lorenz et al. (2016); Nguyen et al. (2017); Ishibashi et al. (2023); Li et al. (2023); Wilson (2024), although Ishibashi et al. (2023) is among the few that also directly considers overfitting in HPO. Their stopping criterion focuses on changes in the expected minimum simple regret, i.e., how much the estimated best objective improves with an additional function evaluation. As with Makarova et al. (2021, 2022), they measure outcomes using RYC and RTC but observe inconsistent results, indicating that while their method can cut computation time, it does not always prevent overtuning.

Fabris and Freitas (2019) conduct experiments with Auto-sklearn (Feurer et al., 2015) across 17 datasets, optimizing the area under the ROC curve. They distinguish among training, internal validation, and external test performance and frequently observe deteriorations from validation to test – phenomena they refer to as "meta-overfitting", especially when datasets are small (around 1000 or fewer observations). Although the validation–test correlation is generally high, the number of SMAC (Hutter et al., 2011) optimization iterations does not correlate with how severe this meta-overfitting is.

Earlier, Escalante et al. (2009) studied a particle swarm optimization (PSO)-based approach to full model selection, including preprocessing, feature selection, learner choice, and hyperparameter tuning. They note that while CV is the main safeguard against overfitting in their experiments, PSO's stochastic exploration can also mitigate the risk of pushing too hard on the validation error. Nonetheless, they acknowledge that repeated exploitation of CV estimates can cause validation improvements not always reflected on a held-out test set.

Lévesque (2018) undertook a large-scale support vector machine (SVM) HPO study with 118 datasets and identify overtuning as a serious problem, especially in small-data scenarios. They test solutions like reshuffling, using an outer test set, and adopting posterior-mean-based selection in BO. Reshuffling helps in small-data regimes – particularly with holdout resampling – while choosing hyperparameters by posterior mean also yields better generalization. Selecting configurations on a separate selection set (Dos Santos et al., 2009; Koch et al., 2010; Igel, 2013), however, can hurt performance because it reduces the data available for HPO. Extending these findings, Nagler et al. (2024) provide a more rigorous analysis of reshuffling, demonstrating its benefits even for simple RS. They further analyze how reshuffling affects the validation loss landscape and derive regret bounds in the asymptotic regime.

Similarly, Larcher and Barbosa (2022) propose dynamic sampling holdout as a faster alternative to CV for AutoML when using population-based algorithms that operate in generations. By reshuffling training and validation partitions at each generation, they reduce the variance and bias inherent in using the same splits repeatedly. Their empirical results show improvements in test performance and lower computational overhead.

Several foundational studies examine the estimation of GE and the variance of GE estimators. A thorough survey by Schulz-Kümpel et al. (2025) benchmarks a broad array of GE confidence-interval construction methods, while earlier and more recent contributions (Stone, 1974; Efron and Tibshirani, 1997; Bengio and Grandvalet, 2004; Austern and Zhou, 2020; Bayle et al., 2020; Bates et al., 2024; Paraschakis et al., 2024) provide theoretical and practical guidance on error estimation. A complementary survey on CV in model selection is offered by Arlot and Celisse (2010).

Empirical comparisons of resampling strategies include Molinaro et al. (2005), who find that in small-sample, high-dimensional genomic studies, naive resubstitution estimates are highly biased, but LOOCV, 10-fold CV, and the .632+ bootstrap can be more reliable. At the same time, the .632+

bootstrap may become biased if the signal-to-noise ratio is high. Further, Wainer and Cawley (2017) systematically evaluate 15 resampling-based HPO techniques for SVMs (with RBF kernels) and suggest that 2-fold or 3-fold CV is often a viable substitute for standard 5-fold CV, providing similar generalization at reduced computational cost. A recent study however finds that current machine learning benchmarks might report the performance of underfitted models due to using an internal holdout procedure instead of cross-validation (Tschalzev et al., 2025), potentially questioning the outcomes of such studies. In clinical prediction models, Dunias et al. (2024) show that standard 5-fold or 10-fold CV tends to yield robust out-of-sample discrimination and calibration, whereas the widely used 1SE rule (Breiman, 1984) can severely miscalibrate predictions in small or low-event-rate samples.

Blum and Hardt (2015) address overfitting to public leaderboards, where participants repeatedly adapt to holdout feedback. They propose the Ladder mechanism, which only reports improvements deemed statistically significant, reducing information leakage and therefore mitigating overfitting. Extending this approach, Hardt (2017) introduce the Shaky Ladder, which adds randomized privacy guarantees so that participants cannot game small improvements. Neto et al. (2016) propose LadderBoot, which injects bootstrap noise to limit the sensitivity of public scores to repeated queries.

Another influential line of work in the context of overfitting in leaderboards leverages differential privacy. Dwork et al. (2015) present Thresholdout and SparseValidate, which provide theoretical generalization guarantees even after multiple adaptive queries to a holdout. Feldman et al. (2019) investigate how easily one can overfit a fixed test set in multiclass settings via adaptively chosen queries. While more classes raise the barrier to overfitting, it remains feasible with relatively few queries. In practice, Roelofs et al. (2019) analyze Kaggle competitions and, surprisingly, detect little evidence of large-scale overfitting, attributing poor generalization more to distribution shifts than to test set overuse.

Arora and Zhang (2021) explore this notion of "meta-overfitting" where continual reuse of a public benchmark – like ImageNet (Russakovsky et al., 2015) – gradually contaminates that benchmark. Researchers copy hyperparameters, architectures, or training procedures that appear to work well on the widely shared test set, therefore implicitly optimizing on it. They propose an information-theoretic approach to quantify how much the test set is effectively "consumed" by repeated usage, suggesting that measuring a model's description length relative to a "pre-test-set" referee can help bound overfitting in such adaptive processes.

Quinlan and Cameron-Jones (1995) point out that more exhaustive searches during rule learning can degrade generalization – a phenomenon they term "oversearching". By fitting random idiosyncrasies in data, broader searches can lead to complex rules that fit the validation set but fail on new data. They propose a layered search strategy that expands search breadth incrementally and stops based on a probabilistic criterion, thereby avoiding the poor test performance often seen with exhaustive strategies.

Similarly, Reunanen (2003) shows that performing CV within variable selection can become self-defeating, because the repeated use of the same data splits to pick features leads to validation overfitting. Following up on this, Reunanen (2007) proposed cross-indexing to avoid overfitting in model selection. Meanwhile, Loughrey and Cunningham (2005) note that aggressive search-based feature selection using methods like genetic algorithms can cause severe overfitting to the validation set, substantially harming test accuracy. They propose an early-stopping mechanism based on CV signals to limit the search depth before overfitting occurs.

Outside of the usage here, the phrase "meta-overfitting" often appears in meta-learning to indicate that knowledge acquired on source tasks may fail to generalize to new, target tasks (Yao et al., 2021; Hospedales et al., 2021; Huisman et al., 2021). It has also been discussed in the context of neural networks (Hospedales et al., 2021), AutoML systems (Yang et al., 2019, 2020), performance prediction (Loya et al., 2023), and other "learning to learn" settings (Barros et al., 2015; Chen

et al., 2023; Song et al., 2024). This differs from the notion of meta-overfitting as used to describe consistent deterioration of test performance relative to validation performance within a single study or single dataset.

In the context of algorithm configuration, which is closely related to HPO, Eggensperger et al. (2019) outline best practices and highlight common pitfalls. They caution that evaluations which are insufficiently diverse, or overly reliant on a small set of training instances, can lead to overtuning, referencing concerns raised in earlier work (Birattari, 2004; Hutter et al., 2007; Birattari, 2009; Hutter et al., 2009). While this prior literature on overtuning in algorithm configuration does not offer a precise formal definition, the phenomenon is generally understood in a manner consistent with our definition in the HPO setting: Overoptimization with respect to few training instances, few random seeds, or specific hardware setups can eventually result in degraded generalization performance on new, unseen test scenarios.

Related, Eimer et al. (2023) point to limited reproducibility in reinforcement learning HPO. Optimizing hyperparameters on very few random seeds often causes severe overfitting, as configurations subsequently do poorly on unseen seeds. The authors advocate for adopting AutoML best practices, such as clear separation of tuning and evaluation seeds and employing systematic HPO strategies.

Many additional studies merely note overtuning or caution against it, especially in small, noisy data (as in certain linear or clinical models (van Calster et al., 2020; Šinkovec et al., 2021; Riley et al., 2021)). While they do not directly measure overtuning, they nonetheless highlight the vulnerability of HPO to misleading improvements when sample sizes are too limited.

Finally, two survey works – Feurer and Hutter (2019) and Bischl et al. (2023) – explicitly identify overtuning as a core problem in HPO. They summarize various strategies for mitigating over-optimization of validation error, referencing much of the research above.

# D Details on an Empirical Analysis of Overtuning

*FCNet* (Klein and Hutter, 2019) is based on exhaustive evaluations of fully connected feed-forward neural networks on four UCI regression datasets: Protein Structure, Slice Localization, Naval Propulsion, and Parkinsons Telemonitoring. Each dataset is randomly split into 60% training, 20% validation, and 20% test sets. The model architecture consists of two hidden layers followed by a linear output layer. For the search space and additional information, see Klein and Hutter (2019). Each configuration (a combination of architectural and training hyperparameters) is trained using the Adam optimizer for 100 epochs, minimizing the mean squared error (MSE), which is also used as the evaluation metric. To account for stochasticity in training, each configuration is repeated four times using different random seeds. This yields a tabular benchmark dataset with complete learning curves and performance statistics for all configurations. We use the final (with respect to the number of epochs trained) validation and test MSE in our analyses. For each replication and dataset combination, we computed the relative overtuning as defined in Definition (3.1) based on an HPC trajectory of all evaluated HPCs ($T = 62208$).

*LCBench* (Zimmer et al., 2021) is based on evaluating 2000 HPCs, sampled uniformly at random, for a funnel-shaped MLP on 35 classification datasets. For the search space and additional information, see Zimmer et al. (2021). Each dataset reserves 33% as a test set, and the remaining data is split into training and validation sets, with the validation set comprising 33%. Models are trained using SGD with cosine annealing (without restarts) and evaluated using accuracy and cross-entropy. We use the final (with respect to the number of epochs trained) validation and test performance values in our analyses. For each dataset and metric combination, we computed the relative overtuning as defined in Definition (3.1) based on an HPC trajectory of all evaluated HPCs ($T = 2000$).

*WDTB* (Grinsztajn et al., 2022) includes different learning algorithms evaluated on a curated benchmark of 45 datasets, categorized into four groups: categorical classification, numerical classification, categorical regression, and numerical regression, where categorical/numerical refers to the feature types. Learning algorithms include Random Forest, XGBoost, Gradient Boosting Tree, ResNet, FT Transformer, SAINT, MLP, and HistGradientBoostingTree. For the search spaces and additional information, see Grinsztajn et al. (2022). For each learning algorithm, RS with approximately 400 HPC evaluations is performed, beginning with a default HPC. The data splitting and evaluation protocol is designed to ensure fair and efficient comparison across datasets: 70% of samples are allocated to the training set (unless this exceeds a predefined maximum), and the remaining 30% is split into 30% validation and 70% test sets, both capped at 50000 samples. The validation set is used exclusively for selecting the best configuration during RS and is distinct from the internal validation set used for early stopping. To adjust for dataset size variability, the number of evaluation folds depends on the number of test samples: one fold for >6000 samples, two for 3000−6000, three for 1000−3000, and five for <1000. All models are evaluated on the same folds to ensure comparability. Performance is measured using accuracy (for classification) and $R^2$ (for regression). In our analyses, we exclude default HPCs and use only the random HPCs. For each learning algorithm and dataset combination, we computed the relative overtuning as defined in Definition (3.1) based on an HPC trajectory of all evaluated HPCs ($T \approx 400$). Figure 1 was created by postprocessing the raw data through a resampling-based simulation. For each dataset-model pair, 100 HPO trajectory replicates were created by subsampling $50\%$ of the (originally drawn uniformly at random) configurations without replacement. Within each replicate, the sequence of validation incumbents was extracted to emulate iterative model selection. These incumbent trajectories were aligned by iteration index and aggregated across replicates. At each iteration, the mean and standard error of both validation and test error were computed, yielding the depicted performance curves with corresponding confidence bands.

*TabZilla* (McElfresh et al., 2023) includes evaluations of various learning algorithms on a total of 176 classification datasets. Learning algorithms include CatBoost, XGBoost, LightGBM, DeepFM,

DANet, FT Transformer, TabTransformer, two MLP variants, NODE, ResNet, SAINT, STG, TabNet, TabPFN (no HPO), VIME, NAM, Decision Tree, KNN, Logistic Regression (Linear Model, no HPO), Random Forest, and SVM. For the search spaces and additional information, see McElfresh et al. (2023). Each dataset uses the ten train/test folds provided by OpenML. Within each training fold, a further split is used to construct a validation set for HPO. The best configuration is selected based on validation performance, and final performance is reported on the test set without retraining. Models are evaluated using accuracy, F1, log loss, and ROC AUC. In our analyses, we exclude runs with fewer than 30 HPC evaluations and exclude the default HPCs, retaining only the random HPCs to be consistent with the other studies that generally rely on configurations sampled uniformly at random. For each learning algorithm, dataset, fold, and metric combination, we computed the relative overtuning as defined in Definition (3.1) based on an HPC trajectory of all evaluated HPCs ($T \approx 29$).

*TabRepo* (Salinas and Erickson, 2024) includes evaluations of 1530 learning algorithm and HPC combinations across 211 classification and regression datasets. We use the "D244_F3_C1530_175" context, restricted to 175 datasets. Learning algorithms include Random Forest, Extra Trees, LightGBM, XGBoost, CatBoost, Linear Model, KNN, and two neural network architectures. For the search spaces and additional information, see Salinas and Erickson (2024). Models are evaluated using 3-fold CV. For each fold, data is split into 90% training and 10% test. All models are trained with bagging, generating out-of-fold predictions for estimating generalization performance. Performance is measured using ROC AUC (for binary classification), log loss (for multi-class classification), and RMSE (for regression). Each algorithm has one default HPC and 200 configurations sampled uniformly at random. In our analyses, we exclude runs with fewer than 30 HPCs. For each learning algorithm, dataset, and fold combination, we computed the relative overtuning as defined in Definition (3.1) based on an HPC trajectory of all evaluated HPCs ($T \geq 30$).

*reshuffling* (Nagler et al., 2024) evaluates four learning algorithms (Elastic Net, Funnel MLP, XGBoost, CatBoost) on ten binary classification tasks, varying the dataset size, resampling strategy, usage of reshuffling, and optimizer. For the search spaces and additional information, see Nagler et al. (2024). A fixed outer test set of 5000 samples is held out and never used during HPO. For HPO, subsets of the remaining data are drawn with training-validation sizes $n \in \{500, 1000, 5000\}$. Resampling strategies include: 80/20 holdout, 5-fold CV, 5x 80/20 holdout, and 5x 5-fold CV, ensuring constant train/validation sizes but varying the splits. Performance is measured via accuracy, log loss, and ROC AUC (BO uses ROC AUC only). The best configuration is retrained on the full HPO data and evaluated on the outer test set. RS is performed with 500 fixed HPCs per replication (ten in total). In our analyses, we use only the RS runs without reshuffled resampling. We revisit BO (HEBO and SMAC for a budget of 250 HPCs) and reshuffling the resampling splits in Section 6. For each learning algorithm, dataset, dataset size, repetition, resampling, and metric combination, we computed the relative overtuning as defined in Definition (3.1) based on an HPC trajectory of all evaluated HPCs ($T = 500$).

*PD1* (Wang et al., 2024) is a large-scale HPO dataset developed for evaluating BO algorithms in deep learning. It consists of 24 tasks, each defined by a dataset (e.g., CIFAR10, ImageNet), a model (e.g., ResNet50, Transformer), and a batch size (determined by hardware). For each task, approximately 500 "matched" and 1500 "unmatched" HPCs are evaluated from a shared four-dimensional search space: learning rate (log scale), momentum (log scale), polynomial decay power, and decay fraction. All tasks use Nesterov momentum with fixed pipelines, varying only optimizer hyperparameters. Each configuration is fully trained and logged with learning curves, including validation cross-entropy loss, error rate, and divergence status. Performance metrics are given by error rate and cross-entropy. We use the "phase1" data (both "matched" and "unmatched"). We exclude ImageNet ResNet50 (all batch sizes), LM1B Transformer (2048), WMT15 German-English xformer (64), and UniRef50 Transformer (128), leaving 18 tasks due to failed/incomplete runs or insufficient full-epoch HPCs or runs where test performance was not available. We use the

final validation and test performances for each task in our analyses. For each task and metric combination, we computed the relative overtuning as defined in Definition (3.1) based on an HPC trajectory of all evaluated HPCs ($T \geq 1300$).

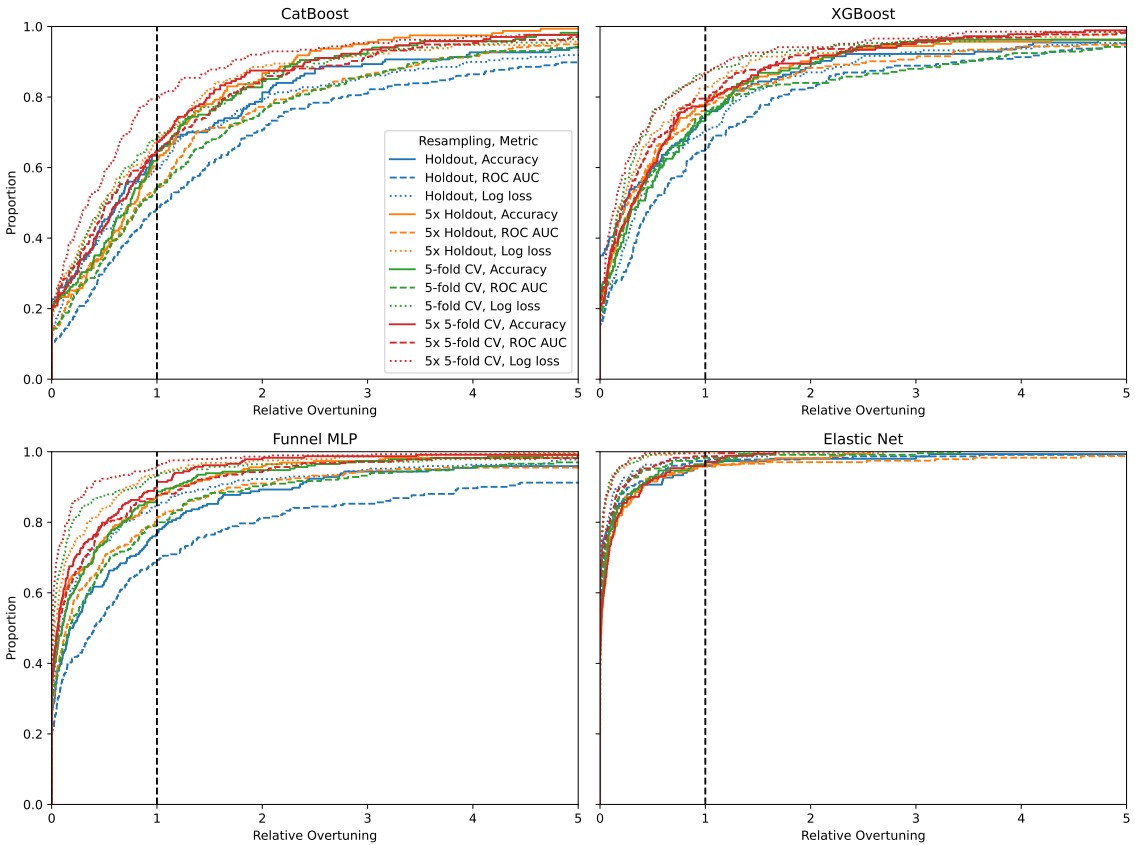

Figure 3: ECDFs of relative overtuning for *reshuffling* (Nagler et al., 2024). Stratified for the learning algorithm, resampling strategy and performance metric but not dataset sizes.

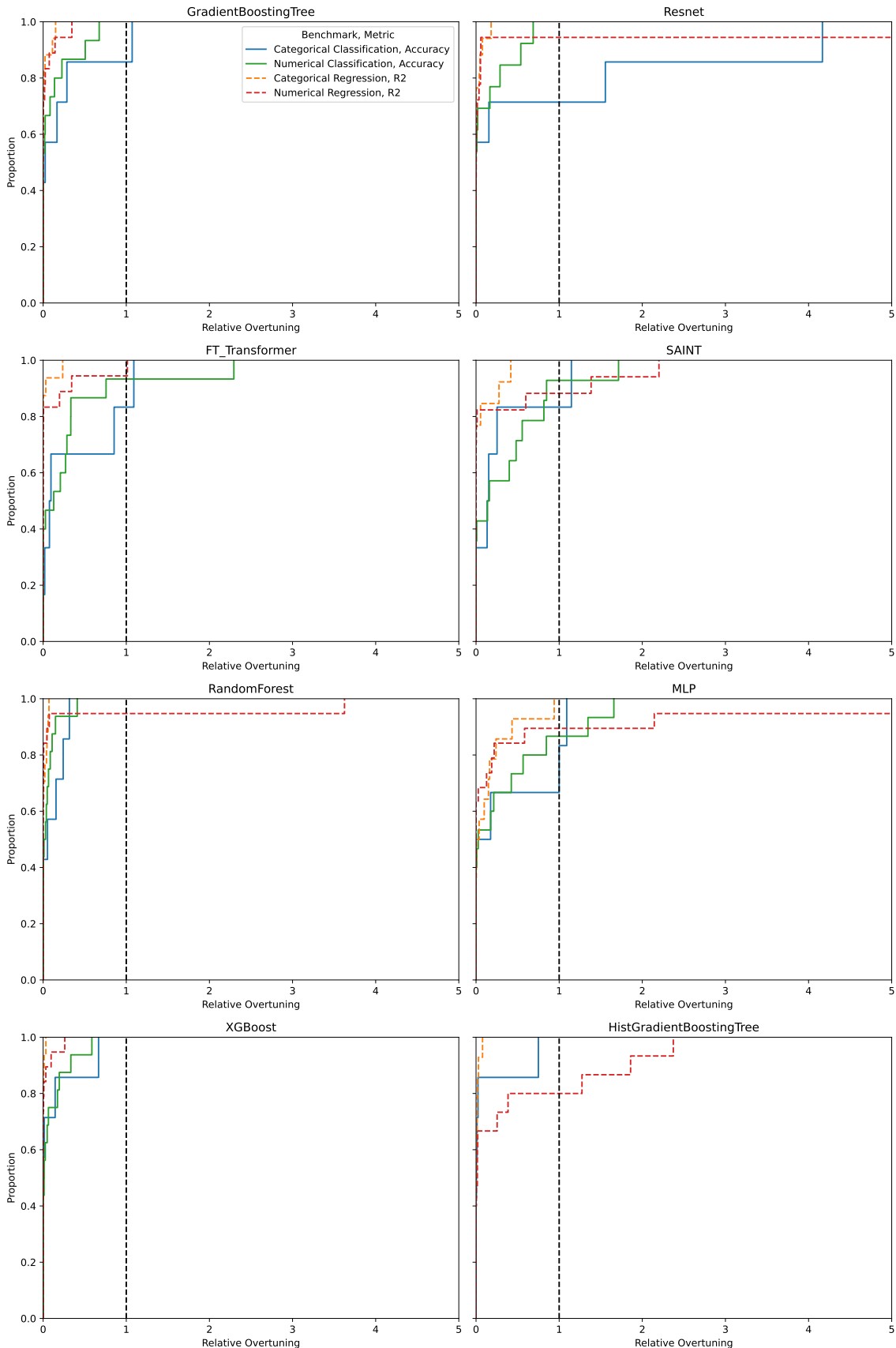

Figure 4: ECDFs of relative overtuning for *WDTB* (Grinsztajn et al., 2022). Stratified for the learning algorithm, benchmark type and performance metric.

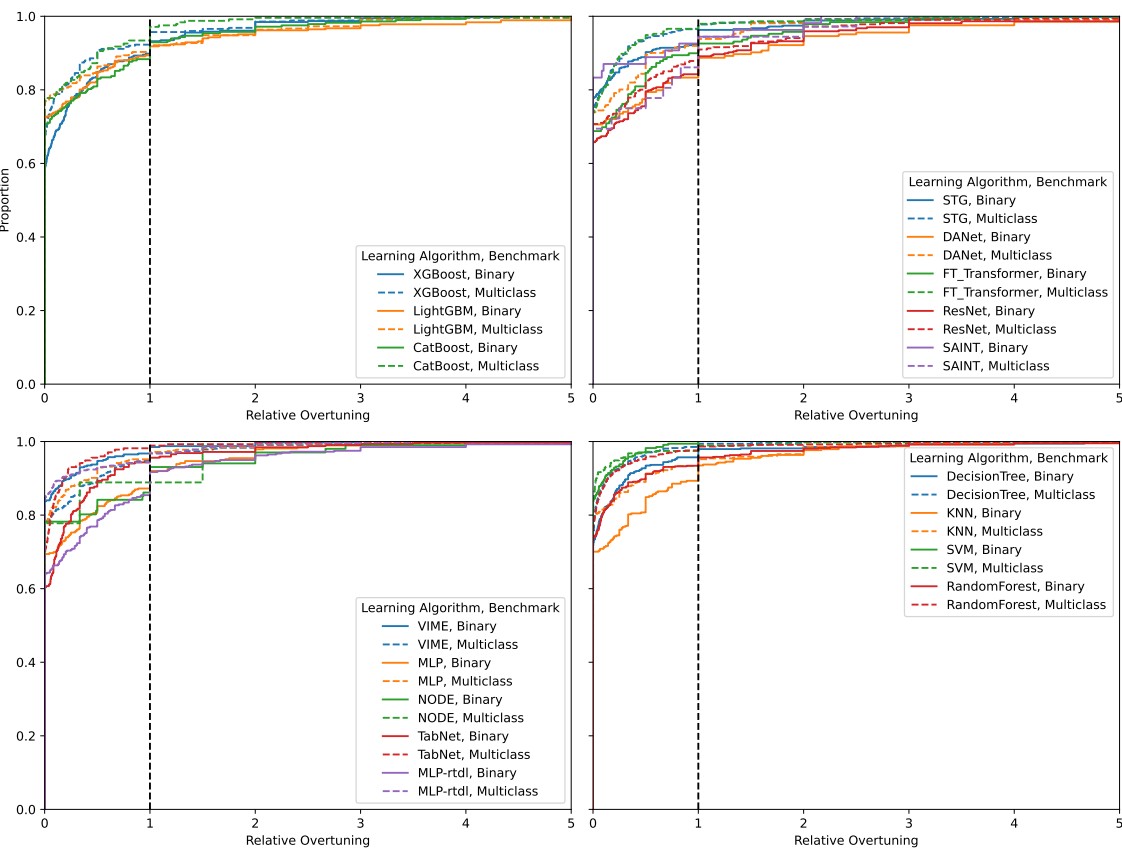

Figure 5: ECDFs of relative overtuning for *TabZilla* (McElfresh et al., 2023). Performance metric accuracy. Stratified for the learning algorithm, and benchmark type.

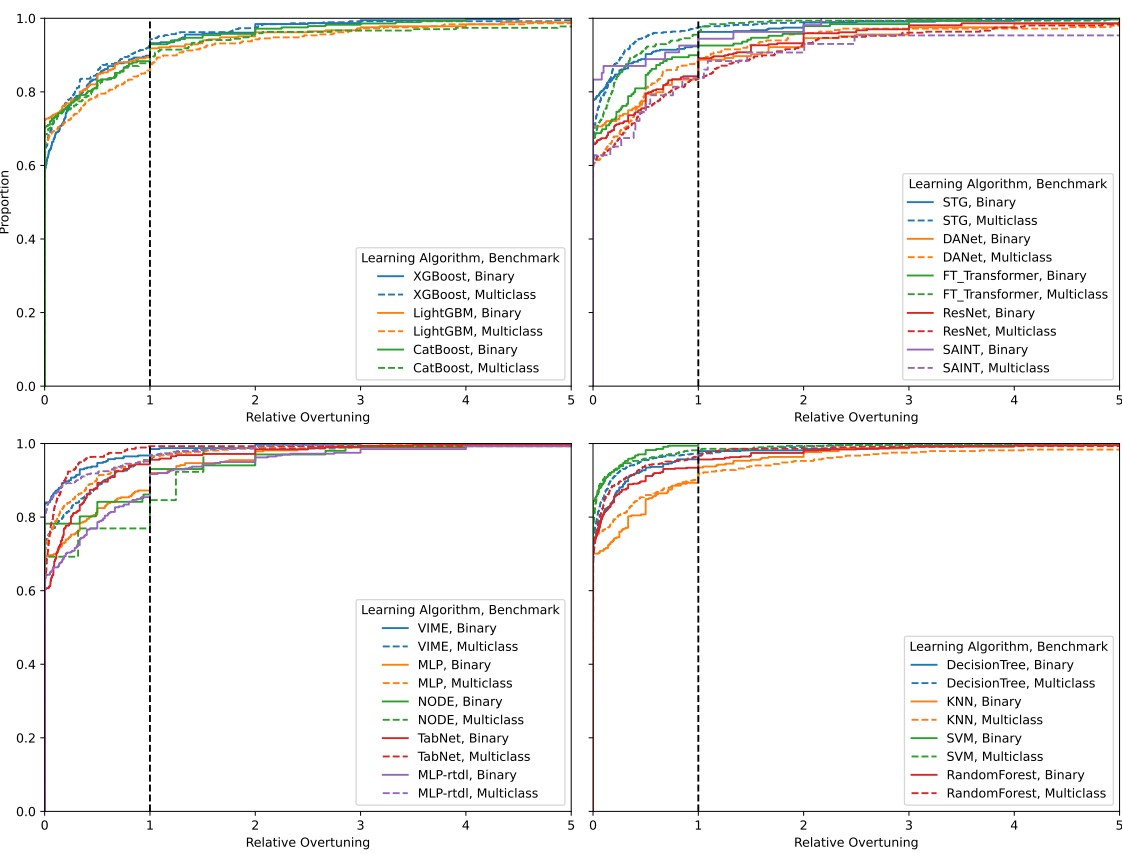

Figure 6: ECDFs of relative overtuning for *TabZilla* (McElfresh et al., 2023). Performance metric F1. Stratified for the learning algorithm, and benchmark type.

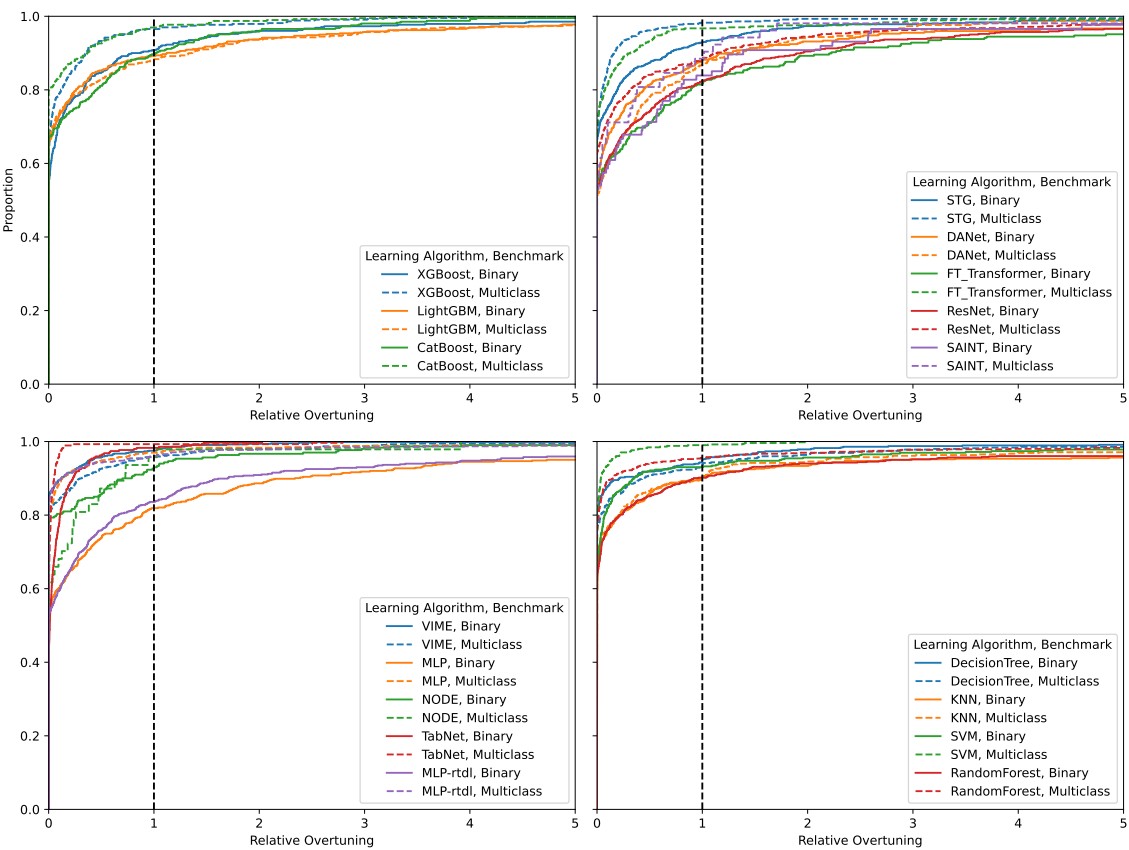

Figure 7: ECDFs of relative overtuning for *TabZilla* (McElfresh et al., 2023). Performance metric log loss. Stratified for the learning algorithm, and benchmark type.

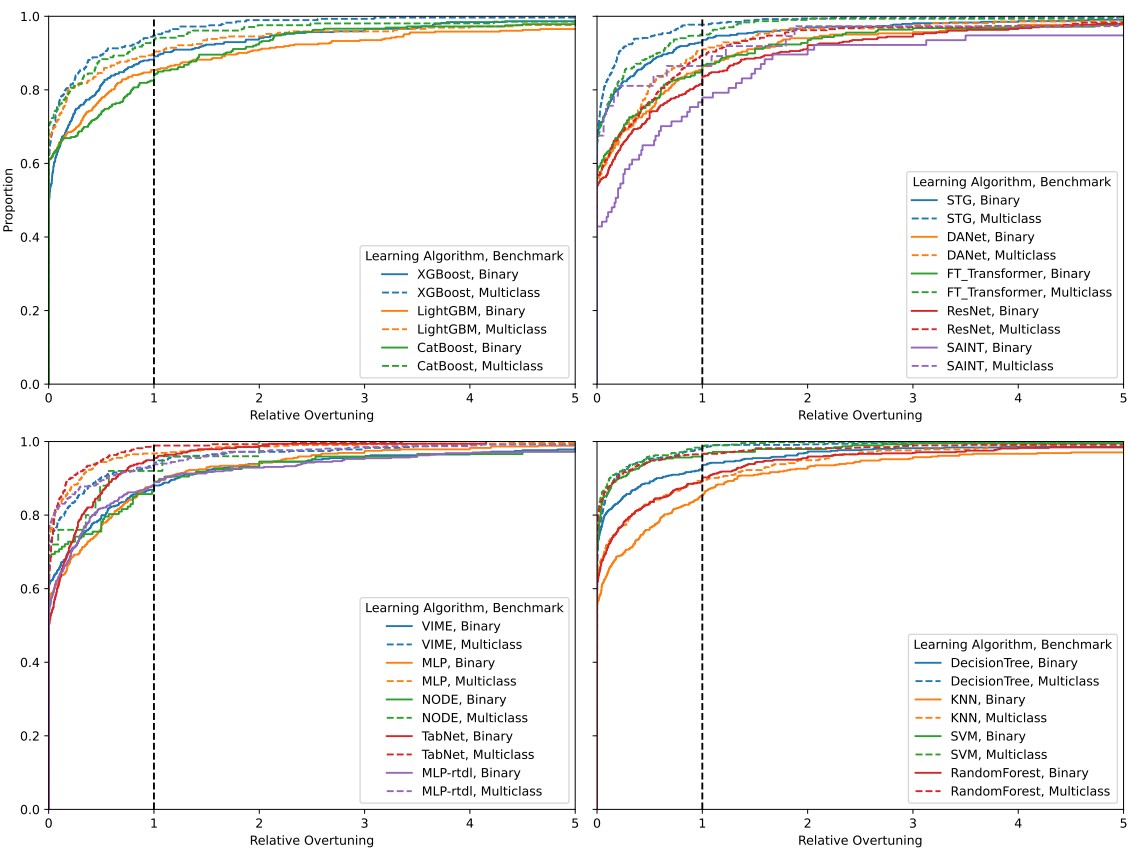

Figure 8: ECDFs of relative overtuning for *TabZilla* (McElfresh et al., 2023). Performance metric ROC AUC. Stratified for the learning algorithm, and benchmark type.

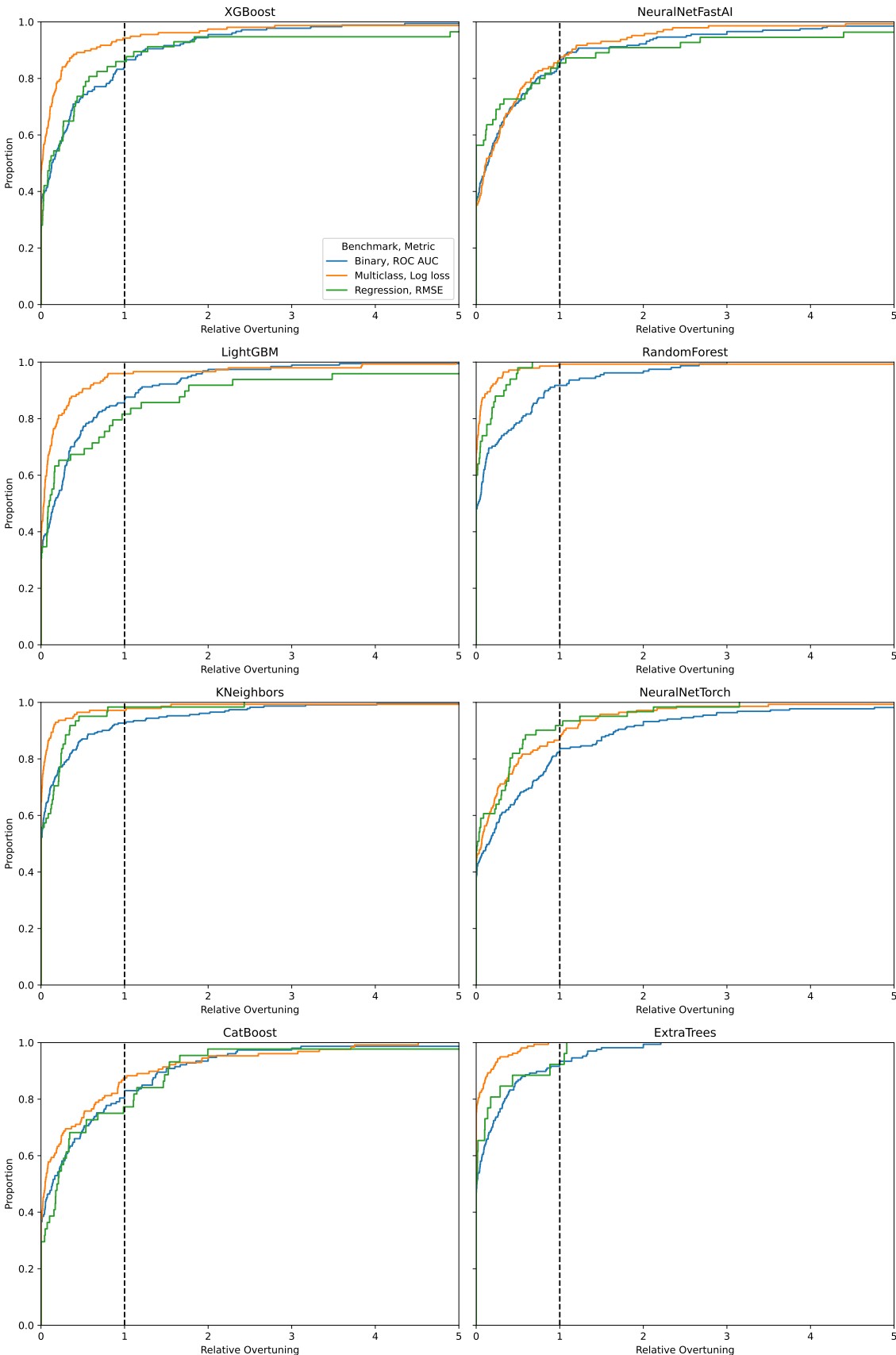

Figure 9: ECDFs of relative overtuning for *TabRepo* (Salinas and Erickson, 2024). Stratified for the learning algorithm, benchmark type and performance metric.

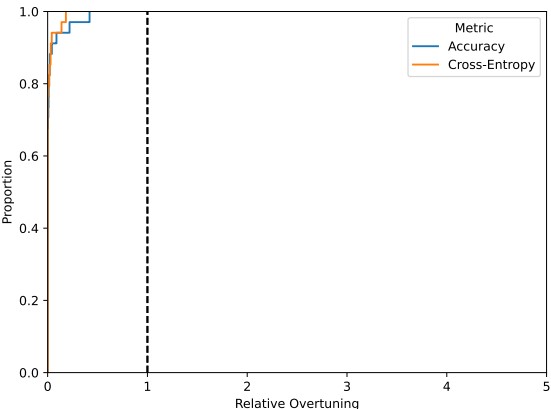

Figure 10: ECDFs of relative overtuning for *LCBench* (Zimmer et al., 2021). Stratified for the performance metric.

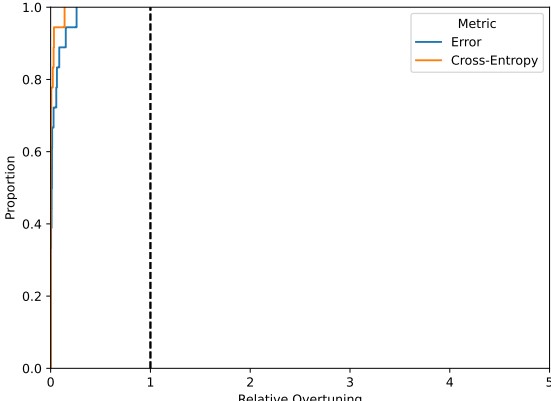

Figure 11: ECDFs of relative overtuning for *PD1* (Wang et al., 2024). Stratified for the performance metric.

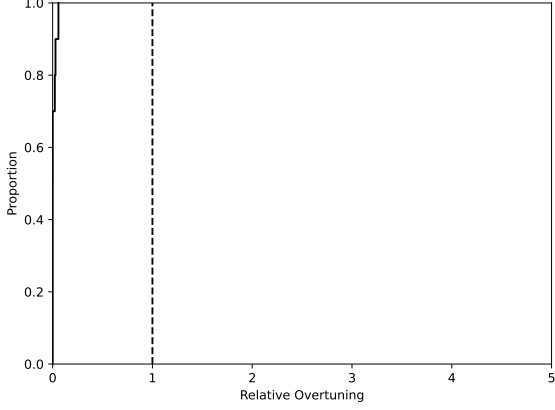

Figure 12: ECDFs of relative overtuning for *FCNet* (Klein and Hutter, 2019).

# E Details on Modeling the Determinants of Overtuning

For an introduction to general linear mixed-effects models, we refer the reader to McCulloch et al. (2008) and Bates et al. (2015). All statistical analyses are interpreted at a significance level of $\alpha = 0.05$. However, we emphasize that we perform many analyses and many of these analyses are conducted on large datasets. As such, statistical significance should be interpreted with caution, as even negligible effects do appear significant due to the large sample sizes. Nonetheless, the magnitude of coefficients as well as associated $z$- and $t$-statistics can still provide meaningful insights into potentially relevant determinants. Finally, we stress that our analysis is exploratory in nature and does not involve the confirmation of pre-specified hypotheses (Herrmann et al., 2024).

(a) Fixed effects results from a GLMM predicting probability of nonzero overtuning.

| Predictor | Estimate | Std. Error | $z$ value | $p$-value |
|---|---|---|---|---|
| (intercept) | -1.161680 | 0.170379 | -6.818 | < 0.001 |
| budget | 3.073490 | 0.054291 | 56.612 | < 0.001 |
| budget$^2$ | -2.034155 | 0.050771 | -40.065 | < 0.001 |
| classifier (CatBoost) | 1.939010 | 0.011118 | 174.395 | < 0.001 |
| classifier (Funnel MLP) | 1.458479 | 0.010659 | 136.827 | < 0.001 |
| classifier (XGBoost) | 1.606069 | 0.010774 | 149.066 | < 0.001 |
| resampling (5x holdout) | -0.300831 | 0.010779 | -27.908 | < 0.001 |
| resampling (5-fold CV) | -0.374434 | 0.010762 | -34.793 | < 0.001 |
| resampling (5x 5-fold CV) | -0.467436 | 0.010747 | -43.493 | < 0.001 |
| dataset size (1000) | -0.290248 | 0.009386 | -30.924 | < 0.001 |
| dataset size (5000) | -0.981109 | 0.009358 | -104.839 | < 0.001 |
| optimizer (HEBO) | 0.083319 | 0.009208 | 9.049 | < 0.001 |
| optimizer (SMAC) | 0.188119 | 0.009245 | 20.347 | < 0.001 |

(b) Fixed effects results from an LMM predicting nonzero relative overtuning on log scale.

| Predictor | Estimate | Std. Error | df | $t$ value | $p$-value |
|---|---|---|---|---|---|
| (intercept) | -1.757e+00 | 1.654e-01 | 1.534e+01 | -10.622 | < 0.001 |
| budget | 3.274e-01 | 5.351e-02 | 1.542e+05 | 6.119 | < 0.001 |
| budget$^2$ | -1.681e-01 | 4.782e-02 | 1.542e+05 | -3.515 | < 0.001 |
| classifier (CatBoost) | 2.151e+00 | 1.136e-02 | 1.542e+05 | 189.408 | < 0.001 |
| classifier (Funnel MLP) | 1.061e+00 | 1.125e-02 | 1.542e+05 | 94.266 | < 0.001 |
| classifier (XGBoost) | 1.554e+00 | 1.143e-02 | 1.542e+05 | 135.941 | < 0.001 |
| resampling (5x Holdout) | -3.350e-01 | 9.392e-03 | 1.542e+05 | -35.666 | < 0.001 |
| resampling (5-fold CV) | -3.544e-01 | 9.428e-03 | 1.542e+05 | -37.594 | < 0.001 |
| resampling (5x 5-fold CV) | -4.969e-01 | 9.478e-03 | 1.542e+05 | -52.423 | < 0.001 |
| dataset size (1000) | -1.973e-01 | 7.924e-03 | 1.542e+05 | -24.905 | < 0.001 |
| dataset size (5000) | -5.188e-01 | 8.519e-03 | 1.542e+05 | -60.901 | < 0.001 |
| optimizer (HEBO) | -3.011e-01 | 8.281e-03 | 1.542e+05 | -36.363 | < 0.001 |
| optimizer (SMAC) | -2.581e-01 | 8.336e-03 | 1.542e+05 | -30.956 | < 0.001 |

Table 3: Fixed effects results of mixed models used to analyze overtuning. BO and RS runs, no reshuffling, test performance of the model retrained on all data. Reference levels: Elastic Net (classifier), holdout (resampling), 500 (dataset size), RS (optimizer).

(a) Fixed effects results from an LMM predicting final meta-overfitting.

| Predictor | Estimate | Std. Error | df | $t$ value | $p$-value |
|---|---|---|---|---|---|
| (intercept) | 5.915e-02 | 1.083e-02 | 9.387e+00 | 5.462 | < 0.001 |
| classifier (CatBoost) | 3.767e-02 | 1.049e-03 | 1.437e+04 | 35.892 | < 0.001 |
| classifier (Funnel MLP) | 2.839e-02 | 1.049e-03 | 1.437e+04 | 27.052 | < 0.001 |
| classifier (XGBoost) | 1.908e-02 | 1.049e-03 | 1.437e+04 | 18.182 | < 0.001 |
| resampling (5x Holdout) | -2.929e-02 | 1.049e-03 | 1.437e+04 | -27.905 | < 0.001 |
| resampling (5-fold CV) | -3.115e-02 | 1.049e-03 | 1.437e+04 | -29.685 | < 0.001 |
| resampling (5x 5-fold CV) | -4.445e-02 | 1.049e-03 | 1.437e+04 | -42.355 | < 0.001 |
| dataset size (1000) | -2.122e-02 | 9.089e-04 | 1.437e+04 | -23.351 | < 0.001 |
| dataset size (1000) | -4.519e-02 | 9.089e-04 | 1.437e+04 | -49.718 | < 0.001 |
| optimizer (HEBO) | 1.623e-03 | 9.089e-04 | 1.437e+04 | 1.786 | 0.074 |
| optimizer (SMAC) | 3.793e-03 | 9.089e-04 | 1.437e+04 | 4.173 | < 0.001 |

(b) Fixed effects results from an LMM predicting final test regret.

| Predictor | Estimate | Std. Error | df | $t$ value | $p$-value |
|---|---|---|---|---|---|
| (intercept) | 2.215e-02 | 4.050e-03 | 9.417e+00 | 5.468 | < 0.001 |
| classifier (CatBoost) | 1.048e-02 | 4.633e-04 | 1.437e+04 | 22.624 | < 0.001 |
| classifier (Funnel MLP) | 1.739e-02 | 4.633e-04 | 1.437e+04 | 37.544 | < 0.001 |
| classifier (XGBoost) | 5.410e-03 | 4.633e-04 | 1.437e+04 | 11.679 | < 0.001 |
| resampling (5x Holdout) | -6.310e-03 | 4.633e-04 | 1.437e+04 | -13.621 | < 0.001 |
| resampling (5-fold CV) | -7.283e-03 | 4.633e-04 | 1.437e+04 | -15.722 | < 0.001 |
| resampling (5x 5-fold CV) | -8.857e-03 | 4.633e-04 | 1.437e+04 | -19.118 | < 0.001 |
| dataset size (1000) | -6.841e-03 | 4.012e-04 | 1.437e+04 | -17.053 | < 0.001 |
| dataset size (5000) | -1.636e-02 | 4.012e-04 | 1.437e+04 | -40.782 | < 0.001 |
| optimizer (HEBO) | -2.073e-03 | 4.012e-04 | 1.437e+04 | -5.167 | < 0.001 |
| optimizer (SMAC) | -2.773e-04 | 4.012e-04 | 1.437e+04 | -0.691 | 0.489 |

Table 4: Fixed effects results of mixed models used to analyze final meta-overfitting and test regret. BO and RS runs, no reshuffling, test performance of the model retrained on all data. Reference levels of factors are: Elastic Net (classifier), holdout (resampling), 500 (dataset size), RS (optimizer).

(a) Fixed effects results from a GLMM predicting probability of nonzero overtuning.

| Predictor | Estimate | Std. Error | z value | p-value |
|---|---|---|---|---|
| (intercept) | -0.368810 | 0.206555 | -1.786 | 0.074 |
| classifier (CatBoost) | 1.740219 | 0.133810 | 13.005 | < 0.001 |
| classifier (Funnel MLP) | 1.813526 | 0.134520 | 13.481 | < 0.001 |
| classifier (XGBoost) | 1.716060 | 0.133593 | 12.845 | < 0.001 |
| dataset size (1000) | -0.376622 | 0.113395 | -3.321 | < 0.001 |
| dataset size (5000) | -1.066268 | 0.114042 | -9.350 | < 0.001 |
| optimizer (HEBO + ES) | -0.549139 | 0.092045 | -5.966 | < 0.001 |

(b) Fixed effects results from an LMM predicting nonzero relative overtuning on log scale.

| Predictor | Estimate | Std. Error | df | t value | p-value |
|---|---|---|---|---|---|
| (intercept) | -1.866e+00 | 2.047e-01 | 3.707e+01 | -9.117 | < 0.001 |
| classifier (CatBoost) | 1.794e+00 | 1.493e-01 | 9.905e+02 | 12.017 | < 0.001 |
| classifier (Funnel MLP) | 6.331e-01 | 1.427e-01 | 9.859e+02 | 4.436 | < 0.001 |
| classifier (XGBoost) | 1.197e+00 | 1.447e-01 | 9.872e+02 | 8.275 | < 0.001 |
| dataset size (1000) | -1.760e-01 | 9.711e-02 | 9.823e+02 | -1.812 | 0.070 |
| dataset size (5000) | -4.765e-01 | 1.065e-01 | 9.870e+02 | -4.475 | < 0.001 |
| optimizer (HEBO + ES) | -2.753e-01 | 8.414e-02 | 9.806e+02 | -3.272 | 0.001 |

Table 5: Fixed effects results of mixed models used to analyze overtuning. BO runs (only HEBO and HEBO with early stopping on 5-fold CV and ROC AUC as performance metric), no reshuffling, test performance of the model retrained on all data. Reference levels: Elastic Net (classifier), 500 (dataset size), HEBO (optimizer). Analyses performed for the final time point which may differ between HEBO and HEBO with early stopping.

(a) Fixed effects results from a GLMM predicting probability of nonzero overtuning.

| Predictor | Estimate | Std. Error | z value | p-value |
|---|---|---|---|---|
| (intercept) | -1.271737 | 0.082493 | -15.416 | < 0.001 |
| budget | 2.239491 | 0.025025 | 89.489 | < 0.001 |
| budget$^2$ | -1.481421 | 0.023734 | -62.418 | < 0.001 |
| metric (ROC AUC) | 0.650435 | 0.004382 | 148.433 | < 0.001 |
| metric (log loss) | 0.295035 | 0.004336 | 68.050 | < 0.001 |
| classifier (CatBoost) | 1.390966 | 0.005180 | 268.533 | < 0.001 |
| classifier (Funnel MLP) | 1.111329 | 0.005116 | 217.220 | < 0.001 |
| classifier (XGBoost) | 1.183944 | 0.005128 | 230.857 | < 0.001 |
| resampling (5x Holdout) | -0.256786 | 0.005042 | -50.934 | < 0.001 |
| resampling (5-fold CV) | -0.302036 | 0.005041 | -59.920 | < 0.001 |
| resampling (5x 5-fold CV) | -0.491480 | 0.005048 | -97.353 | < 0.001 |
| dataset size (500) | -0.256418 | 0.004357 | -58.845 | < 0.001 |
| dataset size (1000) | -0.682647 | 0.004381 | -155.814 | < 0.001 |
| reshuffled (TRUE) | 0.043902 | 0.003555 | 12.351 | < 0.001 |

(b) Fixed effects results from an LMM predicting nonzero relative overtuning on log scale.

| Predictor | Estimate | Std. Error | df | t value | p-value |
|---|---|---|---|---|---|
| (intercept) | -1.563e+00 | 1.375e-01 | 1.689e+01 | -11.367 | < 0.001 |
| budget | 2.604e-01 | 2.952e-02 | 5.379e+05 | 8.819 | < 0.001 |
| budget$^2$ | -1.293e-01 | 2.695e-02 | 5.379e+05 | -4.796 | < 0.001 |
| metric (ROC AUC) | -2.200e-01 | 4.978e-03 | 5.379e+05 | -44.189 | < 0.001 |
| metric (log loss) | -6.831e-01 | 5.118e-03 | 5.379e+05 | -133.462 | < 0.001 |
| classifier (CatBoost) | 2.118e+00 | 6.156e-03 | 5.379e+05 | 344.120 | < 0.001 |
| classifier (Funnel MLP) | 7.089e-01 | 6.082e-03 | 5.379e+05 | 116.555 | < 0.001 |
| classifier (XGBoost) | 1.691e+00 | 6.355e-03 | 5.379e+05 | 266.063 | < 0.001 |
| resampling (5x Holdout) | -2.494e-01 | 5.305e-03 | 5.379e+05 | -47.000 | < 0.001 |
| resampling (5-fold CV) | -2.614e-01 | 5.319e-03 | 5.379e+05 | -49.135 | < 0.001 |
| resampling (5x 5-fold CV) | -4.582e-01 | 5.440e-03 | 5.379e+05 | -84.232 | < 0.001 |
| dataset size (500) | -1.048e-01 | 4.518e-03 | 5.379e+05 | -23.193 | < 0.001 |
| dataset size (1000) | -3.331e-01 | 4.803e-03 | 5.379e+05 | -69.363 | < 0.001 |
| reshuffled (TRUE) | 5.150e-02 | 3.827e-03 | 5.379e+05 | 13.458 | < 0.001 |

Table 6: Fixed effects results of mixed models used to analyze overtuning. RS runs, test performance of the model retrained on all data. Reference levels: accuracy (metric) Elastic Net (classifier), holdout (resampling), 500 (dataset size), FALSE (reshuffled).

(a) Fixed effects results from an LMM predicting final meta-overfitting.

| Predictor | Estimate | Std. Error | df | t value | p-value |
|---|---|---|---|---|---|
| (intercept) | 4.463e-02 | 3.665e-03 | 1.048e+01 | 12.178 | < 0.001 |
| metric (ROC AUC) | 2.973e-02 | 5.864e-04 | 2.877e+04 | 50.697 | < 0.001 |
| metric (log loss) | 1.699e-04 | 5.864e-04 | 2.877e+04 | 0.290 | 0.772 |
| classifier (CatBoost) | 1.398e-02 | 6.771e-04 | 2.877e+04 | 20.648 | < 0.001 |
| classifier (Funnel MLP) | 1.051e-02 | 6.771e-04 | 2.877e+04 | 15.517 | < 0.001 |
| classifier (XGBoost) | 9.060e-03 | 6.771e-04 | 2.877e+04 | 13.381 | < 0.001 |
| resampling (5x Holdout) | -2.839e-02 | 6.771e-04 | 2.877e+04 | -41.928 | < 0.001 |
| resampling (5-fold CV) | -3.596e-02 | 6.771e-04 | 2.877e+04 | -53.119 | < 0.001 |
| resampling (5x 5-fold CV) | -4.633e-02 | 6.771e-04 | 2.877e+04 | -68.421 | < 0.001 |
| dataset size (500) | -1.540e-02 | 5.864e-04 | 2.877e+04 | -26.264 | < 0.001 |
| dataset size (1000) | -3.287e-02 | 5.864e-04 | 2.877e+04 | -56.050 | < 0.001 |
| reshuffled (TRUE) | 1.672e-02 | 4.788e-04 | 2.877e+04 | 34.916 | < 0.001 |

(b) Fixed effects results from an LMM predicting final test regret.

| Predictor | Estimate | Std. Error | df | t value | p-value |
|---|---|---|---|---|---|
| (intercept) | 1.119e-02 | 1.362e-03 | 1.066e+01 | 8.217 | < 0.001 |
| metric (ROC AUC) | 1.095e-02 | 2.742e-04 | 2.877e+04 | 39.950 | < 0.001 |
| metric (log loss) | 1.022e-03 | 2.742e-04 | 2.877e+04 | 3.726 | < 0.001 |
| classifier (CatBoost) | 5.123e-03 | 3.166e-04 | 2.877e+04 | 16.181 | < 0.001 |
| classifier (Funnel MLP) | 1.058e-02 | 3.166e-04 | 2.877e+04 | 33.418 | < 0.001 |
| classifier (XGBoost) | 3.237e-03 | 3.166e-04 | 2.877e+04 | 10.225 | < 0.001 |
| resampling (5x Holdout) | -4.994e-03 | 3.166e-04 | 2.877e+04 | -15.772 | < 0.001 |
| resampling (5-fold CV) | -5.131e-03 | 3.166e-04 | 2.877e+04 | -16.205 | < 0.001 |
| resampling (5x 5-fold CV) | -6.882e-03 | 3.166e-04 | 2.877e+04 | -21.736 | < 0.001 |
| dataset size (500) | -5.088e-03 | 2.742e-04 | 2.877e+04 | -18.554 | < 0.001 |
| dataset size (1000) | -1.030e-02 | 2.742e-04 | 2.877e+04 | -37.571 | < 0.001 |
| reshuffled (TRUE) | -1.529e-04 | 2.239e-04 | 2.877e+04 | -0.683 | 0.495 |

Table 7: Fixed effects results of mixed models used to analyze final meta-overfitting and test regret. RS runs, test performance of the model retrained on all data. Reference levels: accuracy (metric) Elastic Net (classifier), holdout (resampling), 500 (dataset size), FALSE (reshuffled).

(a) Fixed effects results from a GLMM predicting probability of nonzero overtuning.

| Predictor | Estimate | Std. Error | $z$ value | $p$-value |
|---|---|---|---|---|
| (intercept) | -0.903880 | 0.153342 | -5.895 | < 0.001 |
| budget | 2.495057 | 0.092010 | 27.117 | < 0.001 |
| budget$^2$ | -1.626134 | 0.088101 | -18.458 | < 0.001 |
| classifier (CatBoost) | 1.799761 | 0.018983 | 94.808 | < 0.001 |
| classifier (Funnel MLP) | 1.426324 | 0.018159 | 78.545 | < 0.001 |
| classifier (XGBoost) | 1.552415 | 0.018403 | 84.356 | < 0.001 |
| dataset size (500) | -0.049298 | 0.016481 | -2.991 | 0.003 |
| dataset size (1000) | -0.664234 | 0.016094 | -41.272 | < 0.001 |
| reshuffled (TRUE) | -0.264457 | 0.013187 | -20.054 | < 0.001 |

(b) Fixed effects results from an LMM predicting nonzero relative overtuning on log scale.

| Predictor | Estimate | Std. Error | df | $t$ value | $p$-value |
|---|---|---|---|---|---|
| (intercept) | -1.884e+00 | 1.472e-01 | 1.758e+01 | -12.792 | < 0.001 |
| budget | 2.229e-01 | 8.841e-02 | 5.548e+04 | 2.521 | 0.012 |
| budget$^2$ | -1.417e-01 | 8.109e-02 | 5.548e+04 | -1.748 | 0.081 |
| classifier (CatBoost) | 2.080e+00 | 1.842e-02 | 5.549e+04 | 112.902 | < 0.001 |
| classifier (Funnel MLP) | 1.492e+00 | 1.861e-02 | 5.549e+04 | 80.164 | < 0.001 |
| classifier (XGBoost) | 1.710e+00 | 1.920e-02 | 5.549e+04 | 89.042 | < 0.001 |
| dataset size (500) | -6.258e-02 | 1.376e-02 | 5.548e+04 | -4.549 | < 0.001 |
| dataset size (1000) | -3.982e-01 | 1.462e-02 | 5.548e+04 | -27.232 | < 0.001 |
| reshuffled (TRUE) | -2.693e-01 | 1.159e-02 | 5.548e+04 | -23.236 | < 0.001 |

Table 8: Fixed effects results of mixed models used to analyze overtuning. RS runs, subset of runs with holdout and ROC AUC as performance metric, test performance of the model retrained on all data. Reference levels: Elastic Net (classifier), 500 (dataset size), FALSE (reshuffled).

(a) Fixed effects results from an LMM predicting final meta-overfitting.

| Predictor | Estimate | Std. Error | df | $t$ value | $p$-value |
|---|---|---|---|---|---|
| (Intercept) | 8.699e-02 | 2.010e-02 | 9.338e+00 | 4.329 | 0.002 |
| classifier (CatBoost) | 3.000e-02 | 3.055e-03 | 2.375e+03 | 9.820 | < 0.001 |
| classifier (Funnel MLP) | 4.230e-02 | 3.055e-03 | 2.375e+03 | 13.845 | < 0.001 |
| classifier (XGBoost) | 2.066e-02 | 3.055e-03 | 2.375e+03 | 6.761 | < 0.001 |
| dataset size (500) | -4.254e-02 | 2.646e-03 | 2.375e+03 | -16.077 | < 0.001 |
| dataset size (1000) | -9.853e-02 | 2.646e-03 | 2.375e+03 | -37.241 | < 0.001 |
| reshuffled (TRUE) | 5.483e-02 | 2.160e-03 | 2.375e+03 | 25.382 | < 0.001 |

(b) Fixed effects results from an LMM predicting final test regret.

| Predictor | Estimate | Std. Error | df | $t$ value | $p$-value |
|---|---|---|---|---|---|
| (intercept) | 2.330e-02 | 4.770e-03 | 1.022e+01 | 4.885 | < 0.001 |
| classifier (CatBoost) | 9.145e-03 | 1.335e-03 | 2.375e+03 | 6.849 | < 0.001 |
| classifier (Funnel MLP) | 2.310e-02 | 1.335e-03 | 2.375e+03 | 17.303 | < 0.001 |
| classifier (XGBoost) | 7.782e-03 | 1.335e-03 | 2.375e+03 | 5.828 | < 0.001 |
| dataset size (500) | -7.634e-03 | 1.156e-03 | 2.375e+03 | -6.602 | < 0.001 |
| dataset size (1000) | -1.915e-02 | 1.156e-03 | 2.375e+03 | -16.562 | < 0.001 |
| reshuffled (TRUE) | -5.656e-03 | 9.442e-04 | 2.375e+03 | -5.990 | < 0.001 |

Table 9: Fixed effects results of mixed models used to analyze final meta-overfitting and test regret. RS runs, subset of runs with holdout and ROC AUC as performance metric, test performance of the model retrained on all data. Reference levels: Elastic Net (classifier), 500 (dataset size), FALSE (reshuffled).

### E.1 HEBO vs. HEBO with Early Stopping

As mentioned in Appendix A, when HPO protocols follow different search trajectories – due to factors like early stopping, choice of optimizer, or resource constraints – it is necessary to compare the test performance of their incumbents to assess generalization properly since overtuning cannot capture this performance aspect. In Section 6 we have seen that HEBO with early stopping à la Makarova et al. (2022) reduces overtuning in the *reshuffling* study (Nagler et al., 2024) based on 5-fold CV and ROC AUC as performance metric (other factors left at their default, i.e., non-reshuffled resampling and test performance assessed via retraining the inducer configured by a given HPC on all data and evaluating on the outer holdout set). We also visualize this (over all learning algorithms but stratified) for the dataset size in Figure 13 where we observed that HEBO with early stopping indeed exhibits less overtuning.

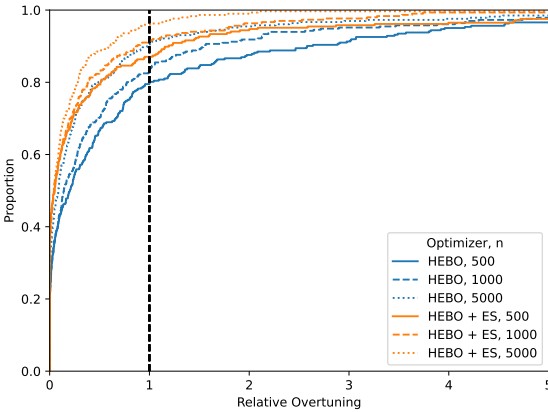

Figure 13: ECDF of relative overtuning for HEBO vs. HEBO with early stopping based on the *reshuffling* study (Nagler et al., 2024). 5-fold CV as resampling. ROC AUC as performance metric.

However, looking at the difference in test performance of the final incumbent returned by HEBO vs. HEBO with early stopping (Figure 14), we observed that HEBO with early stopping does not consistently improve generalization performance. As shown in Figure 14a, HEBO with early stopping yields worse test performance (positive $\Delta$) nearly as often as it yields better test performance (negative $\Delta$) compared to HEBO without early stopping.

To further understand the impact of early stopping on generalization, we analyze the relationship between changes in overtuning and corresponding changes in test performance when comparing HEBO with and without early stopping (Figure 14b). Each point in the scatter plot represents a single HPO run, with the $x$-axis denoting the change in overtuning and the $y$-axis the change in test performance – both computed such that positive values indicate worse outcomes for HEBO with early stopping. We observed a clear positive correlation between the two quantities, suggesting that reductions in overtuning achieved through early stopping tend to coincide with improved test performance. However, this relationship is not uniformly beneficial. While a substantial number of runs fall into the lower-left quadrant, indicating that early stopping reduces overtuning and improves test performance, there are also numerous instances in the upper-right quadrant where early stopping could not decrease overtuning yet harmed generalization (because we stopped too early). Moreover, the majority of points are concentrated near the origin, indicating that early stopping often has only a minor effect. These results confirm that early stopping can mitigate overtuning in some cases, leading to better generalization, but it does not consistently yield improvements and sometimes may even be detrimental.

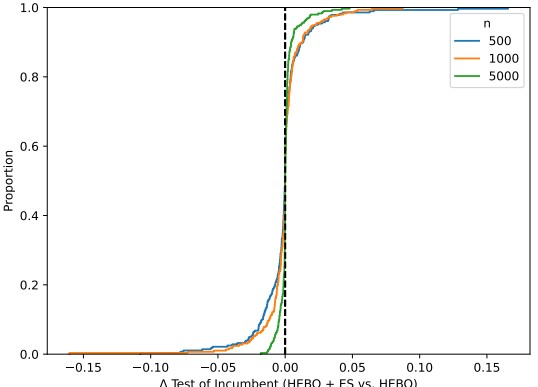

(a) ECDF of the difference in test performance of the final incumbent for HEBO vs. HEBO with early stopping.

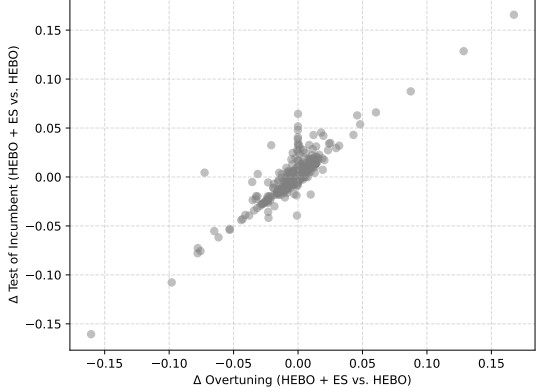

(b) Scatter plot of the difference in test performance of the final incumbent and the differences in final overtuning for HEBO vs. HEBO with early stopping.

Figure 14: Visualizations of the differences in test performance of the final incumbent and the differences in final overtuning for HEBO vs. HEBO with early stopping based on the *reshuffling* study (Nagler et al., 2024). 5-fold CV with ROC AUC as performance metric.

## F Computational Details

As stated in Section 5 and Section 6, we rely on various published works that conducted HPO runs and published this data. With the exception of the HEBO runs with early stopping à la Makarova et al. (2022) as analyzed in Section 6, we did not run any new experiments. For these HEBO runs we used the code base of the *reshuffling* study (Nagler et al., 2024) released under MIT License. Early stopping à la Makarova et al. (2022) was implemented as described in the original paper. We use a patience of 20 HPCs before early stopping can be triggered and use 2000 quasi-random candidate points sampled from the search space to compute the lower confidence bound. We estimate our total compute time for the HEBO with early stopping experiments to be roughly 0.63 CPU years. Benchmark experiments were run on an internal HPC cluster equipped with a mix of Intel Xeon E5-2670, Intel Xeon E5-2683 and Intel Xeon Gold 6330 instances. Jobs were scheduled to use a single CPU core and were allowed to use up to 16GB RAM. Total emissions are estimated to be an equivalent of roughly 345.96 kg $CO_2$. The analyses reported in Section 5 and Section 6 require little computational power and were conducted on a personal computer. We release all our code to perform the analyses reported in Section 5 and Section 6 via `https://github.com/slds-lmu/paper_2025_overtuning` under MIT License.

