# OpenReview forum: "Overtuning in Hyperparameter Optimization"
_automl.cc/AutoML/2025/Methods_Track — AutoML 2025 Methods Track_

### Official Review · Reviewer_7koi · 2025-04-22

**Comments To Authors:**

This paper studies (in great detail) the phenomenon of "overtuning" in the hyperparameter optimization (HPO) literature. Overtuning refers to the concept where the parametrization that is returned by an approach is, in fact, not the one that results in the best performance among the parametrizations that were tested. One reason for overtuning is that other indicators are used for performance assessment that may skew the perception of the actual performance.

The authors re-use result data from established papers in the domain and compute the degree of overtuning as well as various other quantities that suggest when overtuning occurs in the first place. The results indicate that overtuning is mild in most studies, but it does take on a larger role in some instances. The authors also propose some strategies for reducing overtuning.

In general, overtuning is an aspect that does not get that much attention in the community although it is important. Hence, it is good to see that this paper studies it thoroughly. I also appreciate very much that it re-uses the data of prior studies in order to highlight the importance of publishing such data and also to show how relevant overtuning is in the broader sense. The paper is generally excellently written and discusses various different angles about overtuning in great detail. The only downsides I see are that a lot of information is in the appendix and that the general linear mixed-effect models (LMMs, which are greatly utilized to assess various quantities of overtuning) are not really introduced. The latter makes this work less accessible to an audience that is not so familiar with LMMs. The former is acceptable, as the authors try their best to summarize the most important points.

Overall, I think that this paper is a strong submission that should definitely be accepted.

Detailed Comments
-----------------
Although the paper is extremely polished, I have a few minor remarks.

Fig. 1: What ribbons are mentioned in the caption?
49: What are $\mathcal{X}$ and $\mathcal{Y}$? In general, this formalization feels like it is introducing a lot of formalism for the sake of making matters appear rigorous. Many important concepts (such as the loss function and, thus, the generalization error) are just vaguely introduced in normal text and later utilized as if they were defined fully rigorously. For someone who is familiar with more theoretical papers, this feels verbose and superficial. I can imagine though that this is the best practice in this domain, which is acceptable.

**Review Confidence:**

2

**Review Rating:**

9

---

### Official Review · Reviewer_ueLG · 2025-04-29

**Comments To Authors:**

This paper first presents and formalizes the overtuning problem that can arise during the hyperparameter selection phase when buidling a ML algorithm. It discusses the difference between the concept of overtuning and the better-known concept of meta-overfitting.
Experiments are then carried out in an attempt to quantify the pheonomen of hyperparameter overtuning in algorithms that have already been proposed in the literature. The author then constructs machine learning models to predict and try to understand what features appear in the methodologies employed by ML practitioners that may lead to hyperparameter overtunning pheonomen. Finally, the author discusses solutions that could be put in place by the practitioner to reduce these overfitting pheonomen, in order to obtain better overall test performances.

All in all, this paper is very well written and formalized. The angle of approach seems original to me. The paper really brings new elements from a methodological and experimental point of view that can be useful for the ML practitioner, in order to build more reliable models.
I was not aware of the differences between meta-overfitting and overtuning. These notions seem very interesting to me.

I am impressed by all the analyses carried out to try and identify experimentally the source of overtunning phenomena. The methodology employed seems very rigorous. It is interesting to see that some very efficient methods like XGboost or neural networks can lead to such problems. It also confirms that using repeated cross-validation over holdout validation reduce overtuning.

I think this paper will lead to some very interesting discussions at the conference.

I had one small criticism, concerning the explicability of the predictive model to try to deduce the causes of overtuning: the author constructs predictive models in an attempt to identify the factors that explain the overtuning phenomenon. But predicting is not necessarily explaining.  Wouldn't it be better to build a causal model to explain the determinants that lead to overtuning pheonomenes?
Perhaps the factors identified come from selection bias, for example, if all the algorithms that presented overtunning problems had the same methodological problem (for one reason or another). We might then be tempted to identify a cause-effect relationship that doesn't exist if we're just observing a correlation.


Minor comments :

- I found a strange sentence in the abstract : "Given [...] the resampling, we study: Can excessive"
- ligne 246, it should LMMs instead of LLMs, isn't it ? Or I missed something with large language models in this paper.

**Review Confidence:**

3

**Review Rating:**

8

---

### Official Review · Reviewer_kf6P · 2025-05-01

**Comments To Authors:**

This paper defines a set of metrics related to the generalization error of machine learning models whose hyperparameters are tuned using a held-out validation dataset. In particular, they define various metrics related to suboptimal test losses when hyperparameters are set by sequentially minimizing a validation loss. The authors then investigate the presence and degree of what they term overtuning in a selection of previous papers on hyperparameter optimization; they conclude that some amount of overtuning occurs in 40% of the hyperparameter optimization experiments contained in those papers and with a quarter of those constituting "severe" overtuning.

Strengths:
- This paper does an excellent job categorizing and contrasting different suboptimalities in test loss in Section 4
- The statistical tools used to analyze the empirical findings are quite rigorous
- The analysis of the previous work appears to be very thorough and detailed although I am far from an expert in this specific area of the literature
Weaknesses:
- My primary complaint about this work is that it feels rather incremental in the context of previous work: the authors acknowledge that prior works have considered and written about the overtuning effect but have stopped short of formally defining it. Furthermore, their conclusions regarding the mitigation of overtuning are all in line with prior findings: in effect, their discussion in Section 7 could succinctly be summarized as "the things people said worked actually work", which is impressively supported/argued in this work but not particularly revolutionary. Regardless, I do not feel that this should be disqualifying.
- I found the paper at times a bit difficult to read: there were grammatical errors and typos that rose above the level of mere nuisance and actually obfuscated meaning. Minimally, I would encourage the authors to thoroughly revise lines 67-69; that sentence gave me fits to try and parse.

Overall, I found this paper interesting to read and I appreciate the intent behind bringing this overtuning effect to light. It was one of the more unique papers I have read in reviewing for this conference, which made it difficult for me to place on the rating scale below. If pushed to make a binary vote, I would opt for accepting this work but I could easily be convinced otherwise.

**Review Confidence:**

2

**Review Rating:**

6

---

### Meta-Review · Area_Chair_UBL6 · 2025-05-06

**Recommendation:** Accept
**Confidence:** 5

**Metareview:**

I congratulate the authors for having convinced all the reviewers of the significance of the studied subject and the quality of their work. There is strong agreement that the paper should be accepted with some minor comments for final polishing, so I clearly give it a go.